# Parameters for one health genomic surveillance of *Escherichia coli* from Australia

Anne E. Watt [1,2,20], Max L. Cummins [3,4,20], Celeste M. Donato[2,5,20], Wytamma Wirth[2], Ashleigh F. Porter [2], Patiyan Andersson[1,2], Erica Donner[6,7], Australian Pathogen Genomics One Health Working Group*, Amy V. Jennison[8], Torsten Seemann [1,2,5], Steven P. Djordjevic [3,4,21] & Benjamin P. Howden [1,2,5,9,21]

Genomics is a cornerstone of modern pathogen epidemiology yet demonstrating transmission in a One Health context is challenging, as strains circulate and evolve within and between diverse hosts and environments. To identify phylogenetic linkages and better define relevant measures of genomic relatedness in a One Health context, we collated 5471 *Escherichia coli* genome sequences from Australia originating from humans ($n = 2996$), wild animals ($n = 870$), livestock ($n = 649$), companion animals ($n = 375$), environmental sources ($n = 292$) and food ($n = 289$) spanning over 36 years. Of the 827 multi-locus sequence types (STs) identified, 10 STs were commonly associated with cross-source genomic clusters, including the highly clonal ST131, pandemic zoonotic lineages such as ST95, and emerging human ExPEC ST1193. Here, we show that assessing genomic relationships at ≤100 SNP threshold enabled detection of cross-source linkage otherwise obscured when applying typical outbreak-oriented relatedness thresholds (≤20 SNPs) and should be considered in interrogation of One Health genomic datasets.

Understanding the complexity of microbial transmission networks from a One Health perspective has become a priority because of their relevance for the study and management of infectious disease and antimicrobial resistance (AMR)[1]. The formation of the One Health High-Level Expert Panel (OHHLEP) in 2021[2], comprising the Food and Agriculture Organisation (FAO), the World Organisation for Animal Health (WOAH), the United Nations Environment Programme (UNEP), and the World Health Organisation (WHO), signals the need for urgent adoption of One Health principles by public health and research initiatives.

*Escherichia coli*, a well-studied member of the Enterobacterales family, is commonly host-associated while also capable of thriving in diverse environments including water and soil. It is both a common commensal and pathogen, responsible for intestinal and

[1]Microbiological Diagnostic Unit Public Health Laboratory, Department of Microbiology and Immunology at The Peter Doherty Institute for Infection and Immunity, The University of Melbourne, Melbourne, Victoria, Australia. [2]Department of Microbiology and Immunology at The Peter Doherty Institute for Infection and Immunity, The University of Melbourne, Melbourne, Victoria, Australia. [3]Australian Institute for Microbiology and Infection, University of Technology Sydney, Ultimo, New South Wales, Australia. [4]The Australian Centre for Genomic Epidemiological Microbiology, University of Technology Sydney, Ultimo, New South Wales, Australia. [5]Centre for Pathogen Genomics, The University of Melbourne, Melbourne, Victoria, Australia. [6]Future Industries Institute, University of South Australia, Mawson Lakes, South Australia, Australia. [7]Cooperative Research Centre for Solving Antimicrobial Resistance in Agribusiness, Food, and Environments (CRC SAAFE), Mawson Lakes, South Australia, Australia. [8]Public Health Microbiology, Public and Environmental Health, Pathology Queensland, Queensland Department of Health, Brisbane, Queensland, Australia. [9]Department of Infectious Diseases, Austin Health, Heidelberg, Victoria, Australia. [20]These authors contributed equally: Anne E. Watt, Max L. Cummins, Celeste M. Donato. [21]These authors jointly supervised this work: Steven P. Djordjevic, Benjamin P. Howden.*A list of authors and their affiliations appears at the end of the paper. ✉e-mail: steven.djordjevic@uts.edu.au; bhowden@unimelb.edu.au

extraintestinal infections in humans and animals[3–7], respectively referred to as intestinal pathogenic *E. coli* (IPEC) and extraintestinal pathogenic *E. coli* (ExPEC), and a contaminant of agricultural crops and food products[8,9]. Reservoirs of such pathogens have been identified across livestock, wildlife, and the environment[10–12]. Furthermore, while the role of environmental *E. coli* populations in AMR dissemination requires further exploration, waste processing systems expose *E. coli* to pressures that can trigger the uptake of resistance mechanisms[13]. The extensive diversity of *E. coli*, some of which are zoonotic, and their widespread capacity to share mobile genetic elements that carry AMR and virulence gene cargo underline the importance of surveillance of this One Health pathogen.

Over the last decade genomic surveillance has become a common tool for investigating human pathogens and outbreaks[1,14–16], while the WHO's Tricycle project is a key example of a multisectoral surveillance initiative focussed on Extended-Spectrum Beta-Lactamase (ESBL) carrying *Escherichia coli*[17]. In this regard, development of genomics-enabled prospective surveillance systems will be an important step in monitoring the emergence of AMR across sectors. So too will be the establishment of sophisticated workflows for sampling, genomic sequencing and bioinformatic analyses to generate fit-for-purpose microbial genomic surveillance datasets. Currently there is little consensus regarding the approaches required to demonstrate linkages, direct or otherwise, between members of microbial communities co-occurring between distinct hosts and environments. Modern outbreak-related investigations for enteric pathogens increasingly utilises core-genome multi-locus sequence typing (cgMLST) in combination with single nucleotide polymorphism (SNP)-based approaches[18–22], the former being used to rapidly cluster genomes into potentially epidemiologically relevant subsets and the latter generally used for high resolution phylogenetic analysis. However, the utility of such approaches in analysing large-scale, diverse (including spatial, temporal, and host/environment diversity) genomic datasets for evidence of potential cross-source linkage is poorly understood.

Here, utilising a large national multi-sectoral genomic dataset, we explore cgMLST and SNP-based approaches in the identification of cross-source clusters and potential transmission events in *E. coli* and highlight their relevance within a One Health context. Our findings serve as a model for analysis of datasets for *E. coli*, and potentially other Genera, and inform the planning and operation of prospective bacterial genomic surveillance infrastructure nationally and abroad.

## Results

### Collection characteristics
In total, 5471 Australian *E. coli* genomes originating from humans ($n = 2996/5471$, 54.76%), wild animals ($n = 870/5,471$, 15.90%), livestock ($n = 649/5,471$, 11.86%), companion animals ($n = 375/5471$, 6.85%), environmental sources ($n = 292/5471$, 5.33%) and food (289/5,471, 5.28%) were analysed. Collection dates ranged from 1986 to 2022, with 98.32% ($n = 5379/5471$) collected between 2001 and 2022. While most samples originated from Victoria and New South Wales, the dataset contains samples from all states and territories (Fig. 1A). Where such data was available, 76.70% (2298/2996) of human samples were identified as from clinical sources. In total, 52.60% ($n = 1576/2996$) of human samples were sourced from extraintestinal sites, while 32.57% ($n = 976/2,996$) originate from intestinal sites, and 14.75% ($n = 442/2,996$) lacked the appropriate metadata. Among livestock sourced samples, 47.92% ($n = 311/649$) originate from extraintestinal sites, while 35.43% ($n = 230/649$) originate from intestinal sites and 16.64% ($n = 108/649$) unable to be assigned to either category (Fig. 1B).

### Phylogenetics
We identified 8 *sensu stricto* phylogroups within the collection, the most frequently identified being B2 which comprised 39.76% ($n = 2175/5471$) of genomes. Phylogroups A, D and B1 were present at comparable frequencies; 17.46% [$n = 955/5,471$], 14.48% [$n = 792/5,471$] and 12.81% [$n = 701/5,471$], respectively. Most B2 ($n = 1637/2164$; 75.26%) and D ($n = 481/803$; 60.73%) genomes were from humans. While 36.23% ($n = 346/968$) of phylogroup A and 25.39% ($n = 178/701$) of phylogroup B1 genomes were from humans, the majority were sourced from wild animals (A: $n = 213/968$; B1: $n = 213/701$), livestock (A: $n = 231/968$; B1: $n = 84/711$) and environmental (A: $n = 68/968$; B1: $n = 120/711$) sources (Fig. 2).

Phylogroups F, E, G and C were less common (Fig. 2). In total, 827 STs were identified. The collection was phylogenetically diverse (Fig. 2) but predominantly comprised of a small number of STs; while we identified 475 singleton STs, the top thirty STs comprised 61.63% ($n = 3372/5471$) of the collection.

### Identification of STs implicated in cross-source clusters
Within outbreak settings, direct or indirect transmission can be inferred by the concurrence of closely related genomes across different sources in combination with epidemiologically informative metadata. While our dataset lacks the latter, the former may provide sufficient evidence to indicate complex and likely protracted movement of strains in either direction (or potentially both) that can be further investigated. We thus sought insight into potential cross-source genomic clusters observed among the sequence types under analysis. cgMLST was used to determine pairwise allelic distances between genomes and genomic clusters were defined as two or more genomes which exhibit an allelic distance of ≤ 40. In total, 3465 isolates (comprising 151,994 isolate pairs) were identified as putative clusters; the remaining 2006 isolates exceeded this distance from any other isolate in the collection (Fig. 2). Note that clusters in this context does not indicate outbreak associated clusters.

Isolates within ten sequence types (STs 131, 963, 1193, 95, 69, 80, 117, 457, 648 and 57) comprised 35.60% ($n = 1,948/5,471$) of the overall collection (Fig. 2C). Putative cross-source clusters within these sequence types were investigated further, using pairwise SKA Single Nucleotide Polymorphism (SNP) analyses (Fig. 3). A preliminary analysis was used to assess the SNP distance among these STs at which cross-source clusters were identified (Supplementary Fig. 1). Clusters were defined as two or more isolates differing by ≤ 100 SNPs. This cut-off is more conservative than that used to identify putative clusters (≤ 40 allelic distance [cgMLST]) but allows for a greater degree of phylogenetic distance than typical clinically defined SNP distance thresholds. The tested cluster metrics lacked consensus with inconsistent support for the ≤ 100 SNPs cut-off threshold (Supplementary Fig. 5). However, additional supplementary analysis using an additional population clustering framework, PopPunk[23], supported our clustering approach, with all clusters identified using our cgMLST threshold falling within the same PopPunk clusters (Supplementary Information). Of the 92,702 cross-source pairs (pairwise combinations of 1948 strains) analysed, just 2,443/92,702 of pairs (2.65%), comprising 541 strains, exhibited SNP distances ≤ 100 SNPs. The SNP distance thresholds were further classified as follows: > 75 and ≤ 100 ($n = 1014/92,702$; 1.10%; strain count = 474), > 50 and ≤ 75 ($n = 1005/92,702$; 1.09%, 329 strains), or > 20 and ≤ 50 ($n = 421/89,491$; 0.46%; 136 strains). Just three isolate pairs, comprising 5 strains, exhibited SNP distances ≤ 20 SNPs, comprising less than 0.1% of the phylogenetic relationships. Utilising a SNP threshold of ≤ 100 rather than ≤ 20 therefore resulted in an 814-fold increase in identification of cross-source pairs, while still requiring isolates to exhibit a high degree of phylogenetic similarity.

Within the top 10 STs 158 clusters emerged, each comprising two or more isolates differing by ≤ 100 SNPs. Of these, 31 comprised cross-source clusters which had minimum cluster size of 2, a median of 10 and a maximum of 251 (Table 1). Cluster counts varied greatly by ST (Table 1); a network analysis of these clusters, stratified by ST, was then generated to explore the potential interconnectivity of sources (Fig. 4).

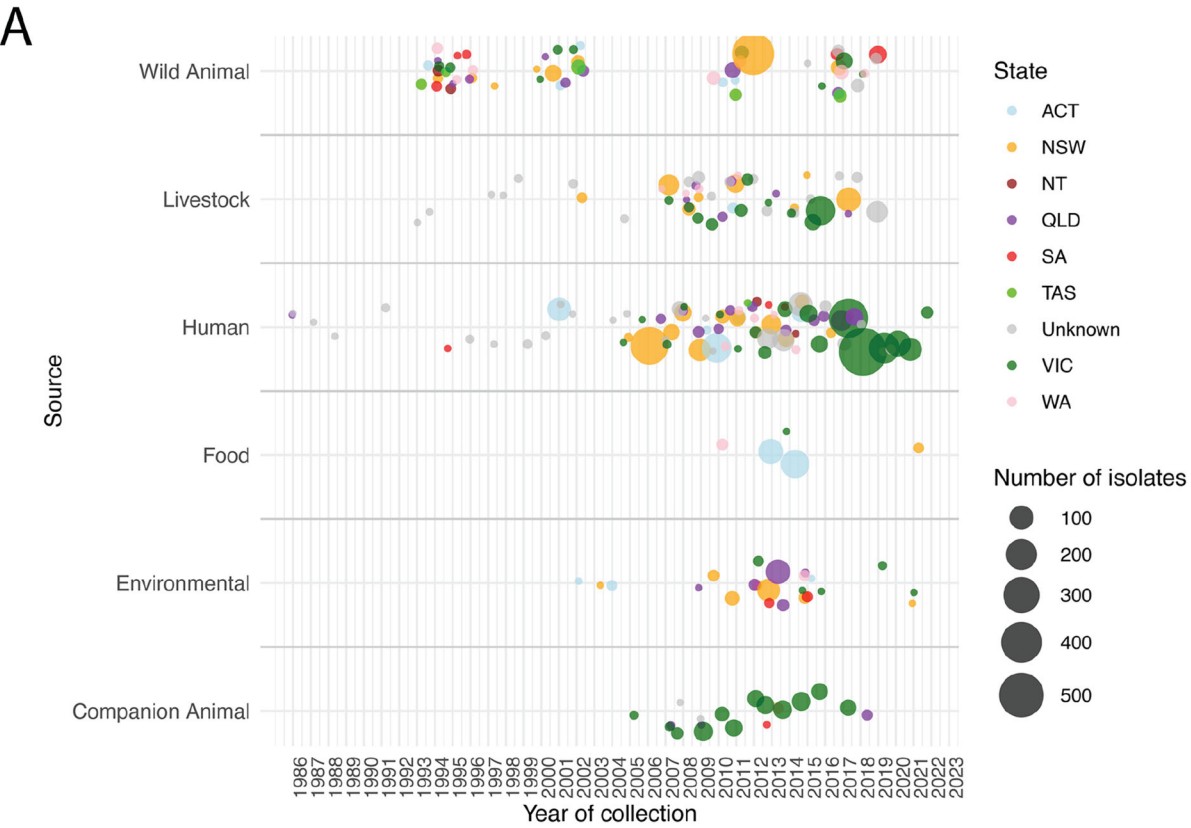

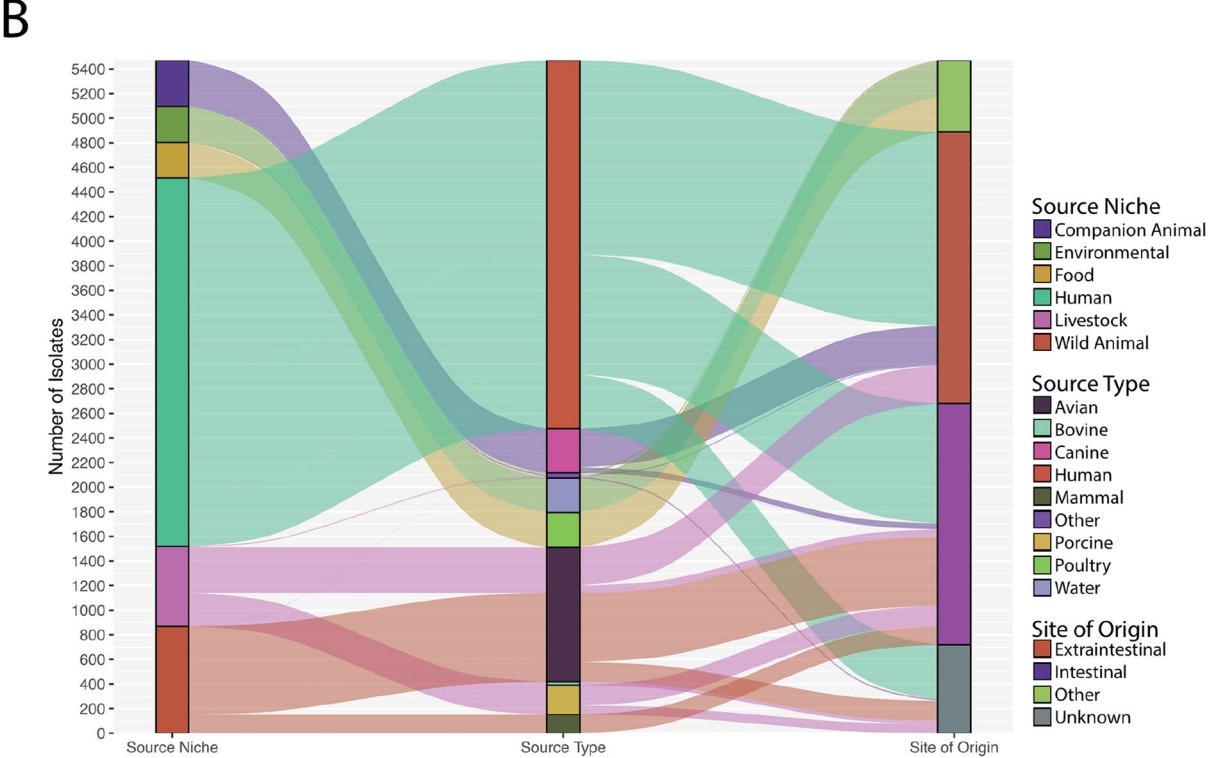

**Fig. 1 | Composition of the collection. A** Timeline of samples in the dataset by source and state or territory of origin. **B** Alluvial diagram visualising the broad source of genomes (left column), species, or type grouping of samples (centre column) and extraintestinal or intestinal sample isolation (right column). Labels for species or type have been omitted for groups with less than 10 samples, these include bovine, food, environmental, soil, marsupial, feline, reptile, caprine and 'liquid'.

Our analysis revealed ST131 was implicated in 10 cross-source clusters spanning humans, companion animals, food, livestock, and wildlife (Table 1). These exhibited a minimum, median and maximum cluster size of 2, 4 and 251, respectively. Additionally, 47 mono-source ST131 clusters were identified among companion animals, food, wildlife and especially humans. Similar trends were also observed with ST1193, which was found to be associated with a total of 9 clusters, of which 7 spanned multiple sources and 2 originated from a single

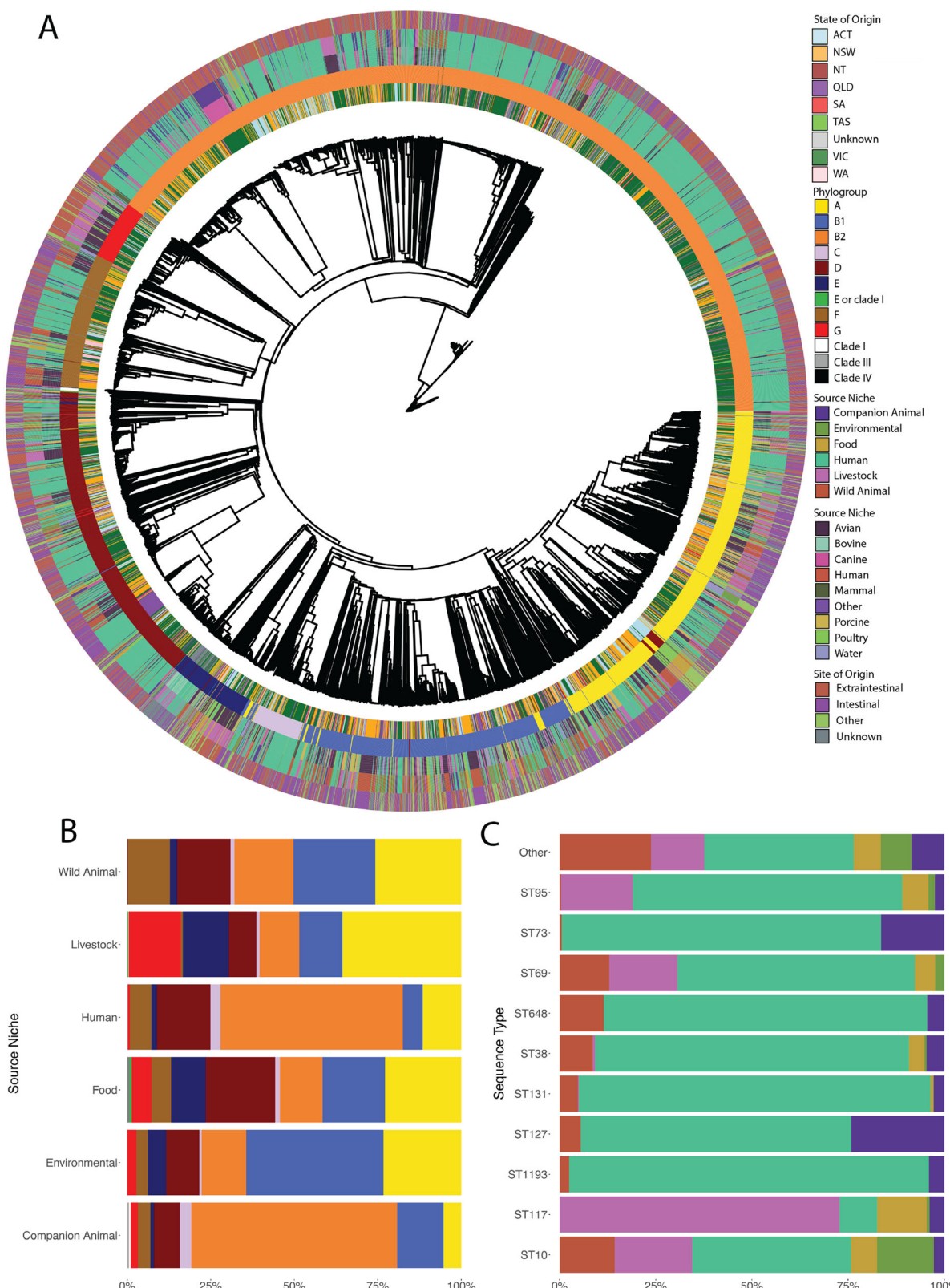

**Fig. 2 | Genomic overview of the collection. A** Phylogenetic tree (cgMLST-derived; see "methods") visualizing the relatedness of the isolates under investigation. Coloured bars indicate (from innermost to outermost) i) the Australian State from which a genome originates; ii) Phylogroup association; iii) Source Niche; iv) Source Type and v) Sample type (whether Extraintestinal, Intestinal, Other or Unknown). Tree is midpoint rooted. Metadata is shown on this tree only to demonstrate its broad associations with phylogenetic structure; data on individual genomes is available in Supplementary Data 1. **B** Intersection of phylogroup and isolate source. **C** Source Niche of the top ten sequence types. See legend to the right of panel A for phylogroup and Source Niche colours associated with panels (**B** and **C**). Note that for panel **B**, cryptic *E. coli* phylogroups are omitted.

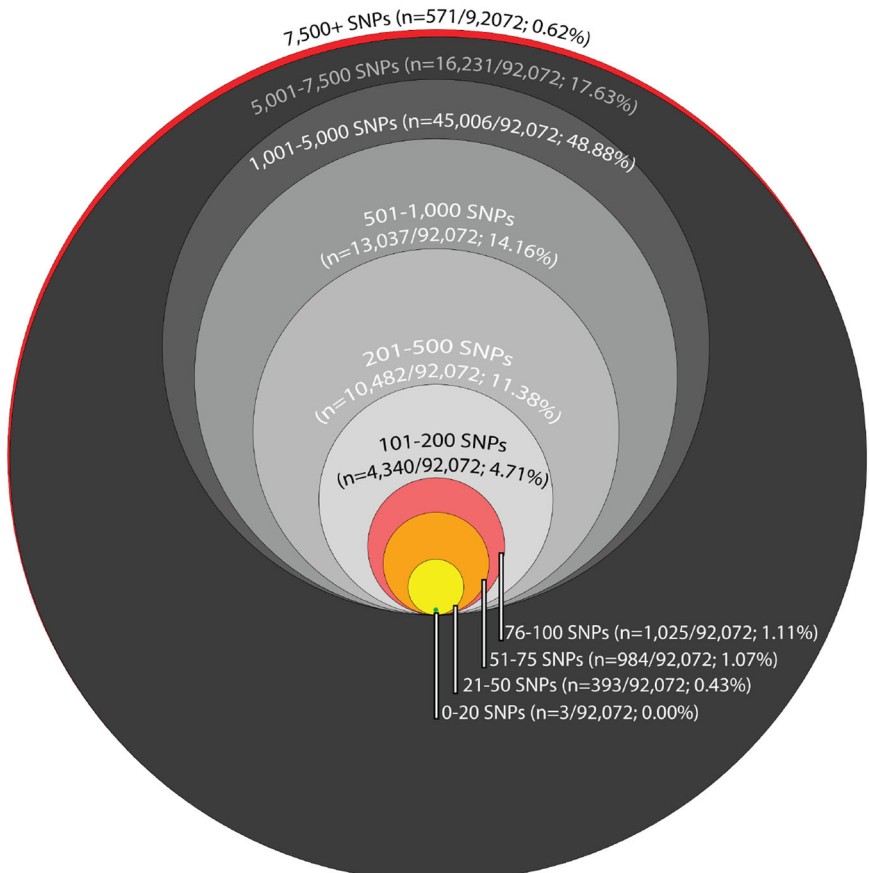

**Fig. 3 | Euler diagram visualising the frequency of SNP distances amongst multi-source isolate pairs.** Area for a given circle is proportional to the number of pairs at a given threshold. The highest distance detected among the ten STs under analysis was 11,350; note that SNP distances were computed only for strain pairs sharing an ST.* - Actual value = 0.03%. Note that large SNP distances (e.g., those exceeding 1000 SNPs) are less accurate than smaller SNP distances and should only be used as a guide.

source. Among ST1193's cross-source clusters, minimum, median and maximum cluster sizes were 20, 68 and 115, respectively. ST963 and ST457 were identified in a total of 3 cross-source clusters (ST963: 1; ST457: 2) spanning wild animals and humans (minimum, median and maximum cluster sizes of: ST963 - 75, 75 and 75; ST457 – 4, 10 and 16, respectively). ST963 was also found to form 2 mono-source clusters, while ST457 was found to form 11. Companion animal associated mono-source clusters were also a feature of networks involving ST131, ST1193 and ST80. Notably, while ST648 exhibited cross-source linkage at ≤ 40 allelic distance, high resolution SNP analyses identified no cross-source genomic clusters at ≤ 100 SNPs.

Four STs, ST117, ST57, ST69, and ST95 featured in poultry-associated networks, some of which were relatively large (Table 1). Regarding ST57, one cross-source cluster spanned food, livestock, and humans, while another spanned food and livestock. ST117 featured predominantly in poultry-associated mono-source clusters, though networks spanning humans, companion animals, food, and livestock were evident. Within ST69, mono-source clusters from poultry and humans predominated, however, cross-source clusters spanning wild animals and humans, and humans and food, were also identified.

We gave particular attention to the cluster network of ST95, i) due to its role as both a human and avian pathogen[24], and ii) as it featured a large cross-source cluster associated with a total of 30 strains, of which 51.6% (n = 16/31) originated from food (poultry meat), 12.9% (n = 4/31) from livestock (diseased poultry animals) and 35.5% (n = 11/31) from humans (Fig. 4A; Supplementary Data 1). Within this cluster (C1, Fig. 4A), the median SNP distance within isolate pairs was 102 (min = 15;

max = 598), while isolate pairs which were collected from different sources exhibited a SNP distance ranging from 33 to 598 (median = 104). In addition, three other clusters were identified spanning multiple sources. The first (C2, Fig. 4A) included 10 isolates, including 2 from food (n = 2/10; 20%) and poultry animals (n = 8/10; 80%), within which SNP distances between isolates collected from multiple sources ranged from 32–53 (median 81). Additionally, Cluster C3 featured 3 isolates from companion animals (canine UTIs) and 7 from humans (including isolates from UTIs and from intestinal biopsies), among which multi-source pairs exhibited SNP distances ranging from 58–94 (median = 83). We also identified large mono-source clusters, including C4, which featured 30 poultry isolates collected from diseased poultry animals between 2014 and 2016 with SNP distances ranging from 0–100 (median = 45), as well as C5, which featured 17 human-sourced isolates collected between 2006 and 2015 with SNP counts ranging from 9–100 (median = 41.5).

**Exploring phylogenetic distance thresholds using cgMLST and SNP-based approaches**

We explored the intersection of cgMLST and SNP-based distance metrics for the 10 STs most commonly associated with cross-source clusters (STs 131, 1193, 95, 57, 69, 117, 963, 457, 80 and 648). Due to the inherent biases within the dataset, we were unable to robustly quantify the correlation of these metrics. Nonetheless, in many instances the metrics correlate and are relatively linear in nature (Supplementary Fig. 3). Notwithstanding sampling bias, some STs appear to have a greater or lesser degree of correlation between these metrics (Supplementary Fig. 3).

**Table 1 | Summary of genomic clusters in SNP-based network analysis**

| ST | Count of ST | Count of Clustered Genomes | Count of Clusters | Mono Sectoral Clusters | Cross Sectoral Clusters | Minimum Cross-Source Cluster Size | Median Cross-Source Cluster Size | Maximum Cross-Source Cluster Size |
|---|---|---|---|---|---|---|---|---|
| 131 | 841 | 610 | 57 | 47 | 10 | 2 | 4.0 | 251 |
| 95 | 247 | 150 | 23 | 18 | 5 | 2 | 10.0 | 31 |
| 1193 | 201 | 153 | 9 | 7 | 2 | 20 | 68 | 115 |
| 69 | 170 | 73 | 19 | 16 | 3 | 3 | 3.0 | 6 |
| 117 | 132 | 89 | 17 | 13 | 4 | 2 | 2.5 | 16 |
| 648 | 113 | 17 | 8 | 8 | 0 | NA | NA | NA |
| 963 | 87 | 80 | 2 | 1 | 1 | 75 | 75.0 | 75 |
| 457 | 69 | 46 | 11 | 9 | 2 | 4 | 10.0 | 16 |
| 57 | 59 | 43 | 8 | 6 | 2 | 10 | 12.5 | 15 |
| 80 | 29 | 19 | 4 | 2 | 2 | 2 | 6.5 | 11 |

An allelic distance threshold of ≤ 40 identified many more strain pairing events than did a SNP distance threshold of ≤ 100 SNPs in isolation (Supplementary Fig. 4). Isolates infrequently exhibited > 40 allelic distance when SNPs were ≤ 100, suggesting a degree of concordance between these methods. However, when visualising the relationship between allelic distance and SNP distance, we found that isolate pairs differing by ≤ 40 alleles differed by as many as 1430 SNPs (Fig. 5). Similarly, SNP counts ≤ 100 were found in isolate pairs which exhibited as many as 51 allelic distances (in which instance the isolate pair exhibited 88 SNPs).

Critically, associations between STs and particular combinations of hosts and environments based on SNP distances were also observed when employing cgMLST allelic distance-based approaches (Fig. 5). For example, STs exhibiting fewer than 100 SNPs exhibited allelic distances between 2 and 20, though isolate pairs which exhibited a greater degree of incongruence between SNP distance and allelic distance were observed; for example, ST131 isolate pairs spanning humans and wild animals exhibited low SNPs and more inflated allelic distances, whilst ST963 isolates from these same sources in some cases exhibited 604 SNPs but an allelic distance of just 4.

### Phylodynamic analysis
We sought to apply phylodynamics methodology to explore cross-source linkage. As temporal data spanned 15 years, sequence types 131, 1193, and 95 were further investigated for suitability for phylodynamic analysis, however, poor temporal signal suggested this approach was not appropriate for this dataset. Root-to-tip regression was used to detect strict clock-like structure in these sequence types which revealed that all three had an $R^2$ below 0.1 suggesting insufficient clock-like behaviour with a strict clock model (ST95 $R^2 = 0.09$, ST1193 $R^2 = 0.02$, ST131 $R^2 = 0.0003$). STs 131 and 95 showed positive correlations and were further investigated for temporal signal using BETS. However, the analysis failed to converge to a stationary distribution, indicating a lack of suitability for phylodynamic analyses.

## Discussion
### Core-genome- and Kmer-based approaches are complementary in large scale One Health genomic surveillance
*E. coli* is the leading cause of death associated with antibiotic resistance[25] and a quintessential One Health pathogen, marking it as an excellent choice for interrogating complex bacterial transmission pathways using whole genome sequencing and AMR[26]. This argument is strengthened by recent studies demonstrating the existence of naturalised environmental *E. coli* populations[27], and evidence that most antimicrobial resistance genes have an environmental origin[28]. Despite this, genomic relationships between *E. coli* across humans,

animals, food, and the environment remain relatively poorly defined, as do the transmission networks underlying these relationships.

Several studies that have sought to understand bacterial transmission networks within a One Health context typically have employed relatedness thresholds that are less conservative than those used in traditional genomic epidemiological investigations of outbreak scenarios[29,30]. There is little consensus on what levels of relatedness are appropriate, partially because of a lack of empirical evidence. Generating such evidence is problematic because of substantial knowledge gaps regarding phylogenetic diversity, and variance in rates of mutation and lateral gene transfer within and between *E. coli* sequence types. Additionally, the impact of changing environmental conditions as *E. coli* moves between hosts and environments remains poorly understood, as it relates to these evolutionary processes. These likely have significant impacts not just on the interpretation of phylogenetic analyses, but for that of phylodynamic investigations which are particularly sensitive to noise introduced by various factors, including but not limited to data associated with large online datasets.

Here we provide novel insight into the intersection of cgMLST-based and SNP-based (SKA) phylogenetic distance measurements within a One Health context to guide future research. While both methodologies are generally in agreement, some isolate pairs appear less closely related based on cgMLST analysis compared with SNP-based analyses suggest and vice versa. This disparity is likely due to the presence of distinct mobile genetic elements between isolate pairs; note that phylogenomic distance using SKA-based methods is performed on a pairwise basis, unlike that of most traditional SNP-based approaches defining a core using pangenome- or reference-based alignments.

We likely underestimate the frequency of cross-source genomic clusters using a SKA-based approach due to the inclusion of SNPs associated with accessory genome content. Future SKA-based methodologies may benefit from removal of sequence data attributed to accessory elements which are likely to inflate SKA-derived SNP values and potentially mask the otherwise close core-genome similarity of genomes under study. It is likely that F plasmids in our study may inflate these SNP values. We recently demonstrated that upwards of 78.6% of a cohort of 34,176 *E. coli* genomes carry an F plasmid representing 1161 plasmid MLSTs[24]. F plasmids are therefore highly phylogenetically diverse and a source of potential SNPs when utilising a method which doesn't mask or omit plasmid sequences. Indeed, preliminary plasmid analyses revealed that at least 76.53% ($n = 4187/5471$) of isolates under analysis carried F-plasmid replicons (Supplementary Data 1). Future studies into F plasmids within a One Health context, and their cross-source and mono-source dissemination, may prove insightful in identifying plasmid lineages which play critical roles in the

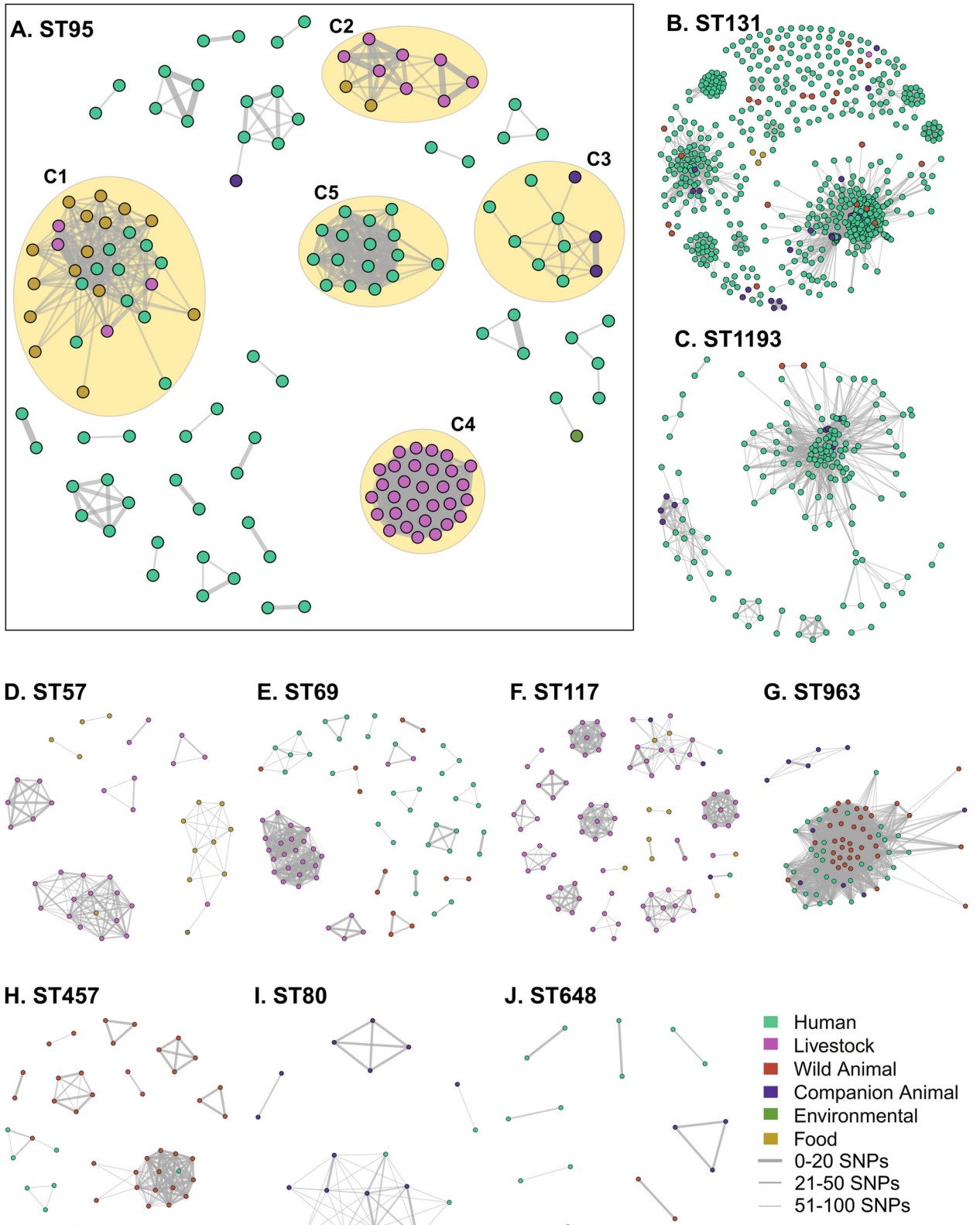

**Fig. 4 | Network analyses for STs commonly associated with multi-sectoral clusters representing their intra-source and inter-source dissemination.** Genomic clusters include two or more instances of genomes which have a pairwise SNP distance of ≤100 SNPs and are coloured by source. **A.** Highlights ST95, a common pathogen of humans and poultry. Examples of multi-source and mono-source clusters are highlighted in yellow. **B–J** Display the networks for other STs commonly identified in the dataset with multi-sectoral clusters. SNP distances for clusters is available in Supplementary Data 2. (https://github.com/maxlcummins/APG-OHEC-Retro-M1/blob/v1.0.1/Supplementary_Material/Supplementary_Table_2.txt).

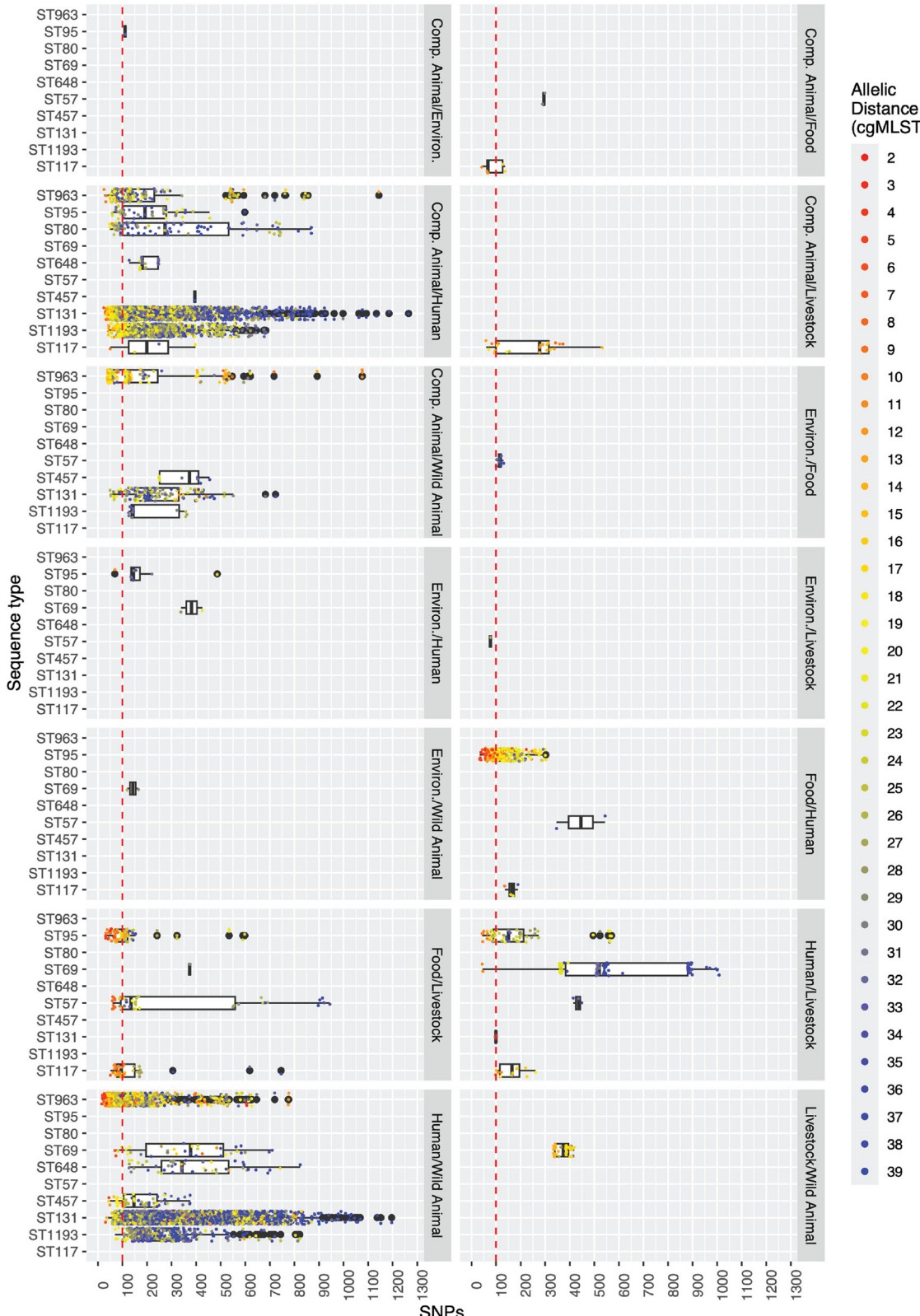

**Fig. 5 | Phylogenetic distance by across sources and sequence types.** This series of box plots compare the phylogenetic distances of isolate pairs (the unit under study). STs for a given pair are shown on the *y*-axis, SNP distances are shown on the *x*-axis and the colour of a data point representing a pair of isolates indicates the cgMLST distance for that pair. Counts of strain pairs for given combinations of STs and pairs of sources are available in Supplementary Data 3. Central lines within boxplots represents the mean, while the bounds of the box indicate the first and third quartiles (25th and 75th percentiles). Whiskers extend to the minimum and maximum values. A red dotted line at 100 SNPs indicates a threshold of relatedness used to indicate moderate phylogenetic overlap. Panels visualise these parameters for pairs of isolates from the denoted pairs of sources – only isolate pairs from different sources are shown.

evolution of virulence and resistance in *E. coli* and other Enterobacterales.

A limitation of our study is that the location of SNPs among closely related genomes is indeterminate. Identifying the location of these mutations (i.e., as being intergenic or intragenic as well as chromosomal or MGE-associated) may provide insight into the mechanisms of host adaptation used by *E. coli* to allow its rapid colonisation (and infection) of a diverse range of hosts. It may also be possible to identify SNPs (or SNP combinations) with predictive power for source attribution, potentially through the use of machine learning, however this is a research area in its infancy[31].

Additionally, due to our approach comparing SNP distances of strains which share a ST, it is likely that we underestimate the frequency of cross-source genomic clusters through omission of genome pairs which may differ by as few as one SNP which has occurred within one of the MLST-defining loci. However, without inferring directionality, our phylogenomic analyses suggest that strains that differ by 50–100 SNPs are occasionally distributed among diverse hosts or over significant geographic distances, reflecting dissemination pathways that may be complex and poorly understood. Knowledge gaps and lack of suitable analytical methodologies for SNP analyses within One Health contexts, as opposed to traditional epidemiological settings, also limit cross-study comparisons. Additional studies are therefore needed to address these gaps in the literature.

Subsequent research exploring phylogenetic distance thresholds within a One Health context should utilise datasets with rigorous sampling methodologies, particularly with genomes from isolates collected from diverse hosts and environments sampled within a fixed geographical and temporal window. We performed a supplementary analysis on a subset of genomes from a five-year window within a single Australian State seeking to identify potential genome pairs which may have been involved in more direct movements between sources due to their more proximal time and area of sampling (See Supplementary Information). While we identified genomic clusters within this period which differ by ≤100 SNPs both within and across sources across multiple sequence types, our sampling depth for any combination of region and sampling window was relatively low.

Snapshot studies focusing sampling efforts on a specific region and time-period would likely provide deeper insight into the relative frequency of cross-source transmission events than we are able to provide with this dataset. To demonstrate this, we also analysed a previously published cohort of 1338 *E. coli* genomes from Nairobi, Kenya (Supplementary Information) which features genomes from isolates collected from humans and their co-residing household livestock as well as rich epidemiologically useful metadata. Within this dataset, we identified clusters of genomes across a diversity of sequence types, which were collected from the same households. Our analysis (and that of the authors) suggests these clusters may be evidence of within-household transmission events.

Future studies should also contrast results from cgMLST and SKA-based approaches with additional reference-based, pangenome-based methods and other high resolution phylogenomic analyses such as wgMLST. We opted for SKA and cgMLST-based approaches due to their scalability and reference-independence. Pangenomic-based analysis methods are highly computationally intensive for large sample sets, while reference-based approaches (e.g., snippy) are influenced by reference bias, and depending on the degree of intra ST diversity, require additional bioinformatic analyses to identify appropriate reference genomes[29]. These methodological considerations are critical for scalable genomic surveillance utilising large datasets. In addition, phylogenetic methods which require distance recalculation for the entire cohort upon addition of new genomes are inherently incompatible for large scale analyses. Different STs will also exhibit a greater or lesser degree of correlation between combinations of cgMLST-based, SKA-based, and potentially other phylogenetic methods. For example, variation in genome diversity and recombination rates within sequence types likely results in cgMLST and SNP distances being differentially impacted by acquisition of accessory gene content, complicating comparisons of these methodologies. Distance thresholds may therefore need to be fine-tuned in a sequence type-specific manner.

## Global pandemic ExPEC sequence types shuttle between humans, animals, food and the environment

Utilising 5471 Australian *E. coli* genomes from humans, animals, food, and the environment spanning 36 years, we have identified *E. coli* clusters spanning every source analysed, including humans, food, companion animals, livestock, wildlife and the environment. Sequence types 131, 963, 1193, 95, 69, 80, 117, 457, 648 and 57 were most frequently associated with cross-source clusters, potentially implicating lineages residing within these sequence types in cross-source transmission. However, it is critical to note that this transmission has likely taken place across unknown periods of time via an indeterminable number of intermediate hosts and vectors. Nonetheless, this data highlights the need for a deeper understanding of: i) relevance of these STs to One Health stakeholders (for example, their zoonotic potential and/or capacity to colonise and persist on food products and man-made surfaces); ii) their carriage of resistance and virulence loci, and the mobile genetic elements that harbour these traits; iii) intra-ST diversity, and; iv) the genomic (and potentially epigenomic) underpinnings of emerging and pandemic STs.

Understanding transmission networks of zoonoses remains a key motivator for One Health genomic surveillance. It is notable that many of the genomic clusters we identified are comprised of pandemic human ExPEC STs[32], including globally disseminated STs with reservoirs in companion animals[33–35] and poultry (Avian Pathogenic *E. coli*)[36]. While human ExPEC are overrepresented in our dataset, and thus closely related strains of major ExPEC lineages are more likely to be identified across sectors, our observations highlight the potential transmission networks which may facilitate their dissemination (Fig. 4) and earmark them for ongoing scrutiny, despite SNP distances within these genomic clusters not being indicative of these being instances of clusters of an outbreak like transmission event. While characterisation of several of these STs has been performed in Australia and abroad[24,37–40], substantial knowledge gaps remain. Several sequence types that were identified in our study, such as in ST69, ST117, and especially ST95, are globally disseminated pandemic avian pathogenic *E. coli* lineages that are increasingly implicated in human extra-intestinal infections[24,32,39]. Our observations suggest that a better understanding of the potential transmission of *E. coli* from all aspects of poultry production (meat and manure) is needed. In addition, protocols are needed to ensure production and application of microbially safe poultry manures for food crop production[41] and use of raw poultry as feed for companion animals. Further, our findings reinforce the need for strict biosecurity in the food production sector to minimise instances of zooanthroponosis ('reverse zoonotic' disease transmission).

Similarly, transmission of *E. coli* isolates between humans and their companion animals (particularly dogs) has been observed in multiple STs, including ST95[42–45]. Clonal isolates of ST131 have also been reportedly isolated from humans and canines in Europe[46]. We also reported previously phylogenetic overlap between isolates of ST131 from humans, canines, wastewater and wild birds which ranged between 37 and 76 SNPs; a trend echoed further in the present dataset[47]. Studies suggest municipal waste and sewage treatment sites visited by wildlife are likely to be important for the movement of ST131 and other *E. coli* STs, as well as other Enterobacterales[38,40,47,48]. We frequently identified genomic clusters of isolates spanning humans and wild animals, particularly wild bird species. Deeper and broader sampling efforts are needed to examine the full extent by which

wildlife and waste streams may enable dissemination of AMR and emerging and pandemic pathogens.

By utilising a relatively high threshold for genomic clusters (100 SNP) we account for protracted transmission pathways potentially involving multiple vectors and reservoirs that would expose *E. coli* to diverse selection pressures[49–51], impacting baseline mutation rates[51] and promoting genetic rearrangements of AMR associated loci[52]. Most studies on *E. coli* mutation rates have occurred in laboratory settings or single hosts[53] and little is known about the impact of diverse selection pressures and transmission across different matrices and lifestyles. Less conservative phylogenetic distance thresholds are therefore justified to gain an understanding of the movement of *E. coli* across sectors and the source combinations and sequence types associated with such movement. Standard clustering metrics were not useful for selecting an appropriate SNP threshold (Supplementary Fig. 5); therefore, we encourage further research on the development and validation of cluster quality indicators that are more suitable for evaluating heterogeneous One Health datasets, such as in the present study.

Here we have collected one of the largest *E. coli* datasets, however, due to the opportunistic sampling and limited metadata associated with this study, our results should be interpreted with the following considerations. Critically, while the current dataset can be used to identify genomic clusters it cannot be used to determine categorically identify transmission events, nor provide insight into the directionality of movement between different sources. Additionally, diverse sampling regimes were utilised across the more than 66 studies from which genomes were aggregated and some sequence types are oversampled. Due to these, and other factors, STs (e.g., ST131) carrying clinically important antibiotic resistance genes predominate in our collection. While we opted not to present AMR-associations in the present work, this predisposition will bias the phylogenetic structure of the cohort being analysed. Similarly, due to the opportunistic nature of the dataset, there are limited spaciotemporal overlaps, potentially reducing the probability of detecting cross-source transmission events. Finally, within the different sectors sampled, particularly livestock and food, sample diversity is restricted, generally being derived from one or two main animal host types (swine and poultry). Despite these limitations, there are valuable signals in a dataset of this size, the use of which is critical for informing the development of future surveillance efforts. Of significance, and despite this sampling bias, we found many mono-source and cross-source clusters across a breadth of STs, shedding light on potential transmission networks.

Future genomic surveillance efforts require rigorous sampling methodologies so that they can provide a basis for a deeper understanding of gold-standard phylogenomic methods within a One Health context. Prospective studies should provide greater insight regarding interhost and mono-source clustering, robustly highlighting mono-source transmission dynamics, such as on-farm disease outbreaks, those in humans in clinical and community settings, or potentially provide evidence of on-farm incursion events of bacterial isolates from wildlife. Phylodynamic methods may be useful in this regard, but they require datasets with large spatiotemporal range; and accumulation of genomic surveillance data over a period of a decade or more at a national scale which may enable identification of incursion events and instances of cross-species transfer in bacterial species as has been demonstrated in viral pathogens such as for canine influenza in the United States[54].

Despite inherent sample bias, genomic clusters of closely related *E. coli* from livestock, food, humans, companion animals and/or wild animals were readily identified, providing evidence of within-host, and cross-source transmission events. Cross-source genomic clusters included pandemic zoonotic ExPEC STs, including ST131, ST95, ST69, ST117 and ST1193. Our findings emphasise the value of adopting One Health approaches to understand bacterial transmission pathways and

pathogen evolution and build towards pathogen genomic surveillance and biosecurity systems in Australia and globally.

## Methods
### Genome collation
We collated all available Australian *E. coli* genomes from in-house collections spanning human, companion animal, livestock, food and environmental sources from the Microbiological Diagnostic Unit Public Health Laboratory (MDU PHL) in Victoria and the Australian Centre for Genomic Epidemiological Microbiology (AusGEM) in New South Wales. The MDU PHL *E. coli* collection includes isolates referred by diagnostic laboratories for different purposes, including suspected carbapenemase-producing *E. coli* (referred for public health purposes under the Public Health and Wellbeing Act 2008, Victoria), clinical isolates submitted for extended antimicrobial susceptibility testing, and *E. coli* isolates derived from primary samples, including environmental and food isolates. All *E. coli* isolates in the collection with successful genomic sequencing (meeting quality control criteria) were included in this dataset under the University of Melbourne ethics approval (Human Research Ethics Committee Reference number: 1954615). Additionally, isolates from the 'Controlling Superbugs' study were also included in this collection (Melbourne Health HREC approval 2013.245). The Australian Centre for Genomic Epidemiological Microbiology (AusGEM) contributed strains sourced from diseased humans[55–57], diseased poultry[58,59], and companion animals[33] as well as healthy swine[60,61] and wild birds[38,62]. Collection of samples utilised in individual research projects was approved by respective ethics panels across collaborating institutions where required (Sydney Local Health District CRGH Human Research Ethics committee [CH62/6/2016-093 – P Chowdhury LNR/16/CRGH/120]; Elisabeth Macarthur Agricultural Institute under the Animal Ethics Committee number [M16/04]; University of New South Wales Animal Care and Ethics Committee [SL101452 14/148 A]). Pathogenic *E. coli* isolates were collected from a breadth of sample types including faecal, urinary and blood specimens from human and animal patients presenting with suspected bacterial infections. In the case of isolates from healthy animal hosts, isolation was from faecal samples and cloacal swabs. As our intention was to collate all available Australian *E. coli* genomes, selection criteria varied between collections aggregated; see individual studies for more detailed sample collection and processing methodology. We also sourced publicly available genome sequences from Enterobase[63] and a 2020 study by Touchon et al.[64], the latter of which was selected due to its publication of a large number of genomes that fit the study criteria. Bioproject accessions for internal collections, as well as metadata and accession numbers for all genomes under analysis, are available for all genomes in Supplementary Data 1. Genomes were required to have metadata detailing year, country, and source of isolation for inclusion.

### Companion scripts
Genomic data was analysed using a custom Snakemake[55] pipeline available at https://www.github.com/maxlcummins/pipelord in all steps except for cgMLST and phylodynamic workflows (described below). Scripts used in the processing and visualisation of data are publicly available on Github (https://github.com/maxlcummins/APG-OHEC-Retro-M1). Default parameters were used unless otherwise stated.

### Genome pre-processing
Read sets were filtered and trimmed using fastp v0.20.1 and assembled using Spades[56] v3.14 via shovill (www.github.com/tseeman/shovill) v1.0.4. Quality control ensured genomes exhibited: i) a length of between 3,800,000 and 6,615,000 bases using assembly-stats (https://github.com/sanger-pathogens/assembly-stats); ii) a known or novel multi-locus sequence type (MLST) of *E. coli* (Achtman scheme) (www.github.com/tseeman/mlst, v2.19.0); iii) 50% of sequence reads

mapping to *E. coli* using Kraken2[57] v2.1.2 (parameter: '--db bacteria') and Bracken[58] v2.5; iv) less than 10% contamination and greater than 90% completeness scores using CheckM[59] v1.2.0. Isolates were also required to have 95% of 2,513 cgMLST alleles present to be considered for further analysis.

## Phylogenetic analysis

Core-genome multilocus sequence types (cgMLSTs) were generated using chewBBACA[60] v2.8.5 (cgMLST schema https://enterobase.warwick.ac.uk/schemes/Escherichia.cgMLSTv1 - accessed 19/07/2022) via the wrapper Coreugate (https://github.com/MDU-PHL/Coreugate) v2.0.5, the latter generated pairwise allelic distances and a phylogenetic tree using rapidNJ[61] v2.3.3 using default settings. When ≥ 2 isolates exhibited an allelic distance ≤ 40, they were considered putative genomic clusters. Consensus on appropriate thresholds for One Health microbial genomic epidemiology are lacking, however a threshold of ≤ 100 allelic distances has been used for preliminary clustering in other research exploring One Health transmission events[26]. We opted for a more conservative threshold of ≤ 40 to minimise high-resolution comparison of genomes differing by large numbers of SNPs. Due to the volume of genomic data under analysis, this preliminary screening step reduced the need for an "all vs all", high-resolution phylogenetic analysis, allowing us to focus our efforts (and computational processing power) on a subset of genomes of interest. We also performed an additional preliminary clustering analysis using PopPunk[23] v2.67 using default settings and the v2 *E. coli* database (available at https://ftp.ebi.ac.uk/pub/databases/pp_dbs/escherichia_coli_v2_full.tar.bz2).

Subsequent, high-resolution phylogenetic analyses on a subset of 10 STs most commonly associated with putative genomic clusters, on a ST-wise basis, was performed using Split Kmer Analysis (SKA)[62] v1.0, with two or more strains differing by ≤ 100 SNPs considered a genomic cluster and any individual two strains meeting this criteria being classified as a clustering pair. This threshold was chosen after a thorough search of the literature; few studies have assessed one health transmission as opposed to direct transmission, and there was little concordance in examining cross sectoral relationships within *E. coli*. Thresholds ranged from 15 to 100 SNPs with clonality considered up to 200 + SNPs[63–67]. As the aim of this study was to examine One Health genomic clusters and not direct transmission, a threshold of ≤ 100 SNPs was chosen.

A core-genome alignment was generated for ST131, ST1193 and ST95 using snippy[68] v4.4.3 and a maximum-likelihood tree was built using IQ-Tree[69] v2.0.3 (parameters: '-m GTR + F -bb 1000'). These sequence types were selected for analysis given their relatively high frequency and implication with cross-source genomic clusters. Core genomic alignments for the three sequence types (STs) were assessed, in combination with their collection years, for temporal structure (i.e., clock-like mutation rates) using both TempEst[70] v1.5.3 and Bayesian Evaluation of Temporal Signal (BETS)[71]. Phylogroups were assigned using Clermontyper[72] v2.0.3.

## Statistical analysis and visualisation

Statistical analyses were performed using R version 4.0.2. Packages utilised for statistical analysis and general processing and visualisation in R include: igraph v2.0.3[73], tidyverse[74] v2.0.0, ggplot2[75] v3.4.4, ggrepel[76] v0.9.3, ggalluvial[77] v0.12.5, ggvenn[78] v0.1.10, plotly[79] v4.10.2, caret[80] v6.0-94, pheatmap[81] v1.0.12, circlize[82] v0.4.15, and scales[83] v1.3.0. Non-parametric correlation analyses for SNP and cgMLST distances were performed using the Spearman Correlation coefficient via ggpubr[84] v0.6.0. Cluster metrics, including Silhouette, Calinski-Harabasz, Davies-Bouldin, Within-Cluster Sum of Squares, Cohesion, and Separation, were used to analyse SNP cutoff thresholds. The analysis was performed in Python version 3.10.12 using sklearn[85] v1.2.2, scipy[86] v1.11.4, and networkx[87] v3.3. The change in each cluster metric due to different cut-off thresholds was statistically assessed using a permutation test with 1000 iterations.

## Ethics

Ethical approval was received from the Royal Melbourne Hospital Human Research Ethics Committee (study number RMH83761). All authors verify the integrity and completeness of data and analyses.

## Reporting summary

Further information on research design is available in the Nature Portfolio Reporting Summary linked to this article.

## Data availability

Metadata and accession numbers are available for all genomes in Supplementary Data 1.

## Code availability

Scripts used in the processing and visualisation of data are publicly available on Github[88] (https://github.com/maxlcummins/APG-OHEC-Retro-M1), as are pipelines used for genomic analysis[89] (https://github.com/maxlcummins/pipelord).

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

## Acknowledgements

This work was funded by the National Health and Medical Research Council (NHMRC) through the Medical Research Future Fund (MRFF): the AusPathoGen Programme (FSPGN000049), as well as Australian Centre for Genomic Epidemiological Microbiology (AusGEM), a collaborative research initiative between the New South Wales Department of Primary Industries and the University of Technology Sydney. The Melbourne Genomics Health Alliance (funded by the State Government of Victoria [Australia]) supported the 'Controlling Superbugs' study. BPH is supported by an NHMRC Investigator Fellowship (GNT1196103). We would also like to thank Tom Hadzioannou for his contribution to ensuring code reproducibility and maintenance.

## Author contributions

B.H., S.D., A.W., M.C. designed the project. BH and SD secured funding. W.W., P.A., C.D., E.D., T.S. and A.J. contributed to project design and analytical approaches. A.W., M.C., T.S., W.W. and A.P. analysed the data. B.H. and S.D. provided data. A.W. and M.C. draughted the manuscript with C.D., S.D. and B.H. All authors reviewed and approved the manuscript.

## Competing interests

The authors declare that there are no conflicts of interest regarding the generation and publication of this manuscript.

## Additional information

## Australian Pathogen Genomics One Health Working Group

Vitali Sintchenko[10], Alicia Arnott[11], Alireza Zahedi[8], Rowena Bull[12], Jessica R. Webb[2], Danielle Ingle[2], Kristy Horan[2], Tuyet Hoang[2], Angeline Ferdinand[2], Tehzeeb Zulfiqar[13], Craig Thompson[8], Lex E. X. Leong[14], Bethany Hoye[15], Glenn F. Browning[16], Michelle Wille[2], Rose Wright[17], Angela Donald[2], Zoe Bartlett[18], Avram Levy[19], Christina Bareja[17], Tatiana Gonzales[17], Cara Minney-Smith[19], Erin Flynn[14], Aruna Phabmixay[17] & Thy Huynh[17]

[10]Centre for Infectious Diseases and Microbiology Laboratory Services, Institute of Clinical Pathology and Medical Research, NSW Health Pathology, Westmead Hospital, Westmead, New South Wales, Australia. [11]Victorian Infectious Diseases Reference Laboratory, Royal Melbourne Hospital, Doherty Institute for Infection and Immunity, Melbourne, Victoria, Australia. [12]School of Biomedical Sciences, Faculty of Medicine, University of New South Wales, Sydney, New South Wales, Australia. [13]National Centre for Epidemiology and Population Health, The Australian National University, Canberra, Australian Capital Territory, Australia. [14]SA Pathology, Adelaide, South Australia, Australia. [15]Environmental Futures, School of Earth, Atmospheric and Life Sciences, University of Wollongong, Wollongong, New South Wales, Australia. [16]Asia-Pacific Centre for Animal Health, Melbourne Veterinary School, Faculty of Science, University of Melbourne, Parkville, Victoria, Australia. [17]Office of Health Protection, Australian Government Department of Health and Aged Care, Canberra, Australian Capital Territory, Australia. [18]Food Standards Australia and New Zealand, Canberra, Australian Capital Territory, Australia. [19]Department of Microbiology, PathWest Laboratory Medicine WA, Queen Elizabeth II Medical Centre, Perth, Western Australia, Australia.

