## [Peer Review file · Nature Communications]

Parameters for One Health genomic surveillance of *Escherichia coli* from Australia

Corresponding Author: Professor Benjamin Howden

Version 0:

Reviewer comments:

Reviewer #1

(Remarks to the Author)

Summary:

The authors of this study assembled a large and diverse collection of *E. coli* genomes from multiple different sources spanning 36 years in Australia to better understand how the different environments are linked with regards to transmission of *E. coli* isolates. They compared cgMLST and SNP based approaches to cluster the datasets into different groups of related isolates and identified the shared sources they were found in. One Health approaches are gaining a lot of attention with regards to control of AMR and it is important to define transmission of *E. coli* on a large scale using whole genome sequencing.

Major comments:

While the authors used a thorough method to define clusters of related isolates, tools for clustering genomes do exist and have been previously used to define clusters in One Health studies. For example, PopPUNK. I'm wondering how the results using this tool would compare to those reported in this study?

Although it was evident from this study that different STs/groups of genetically distinct isolates can persist in multiple environments, there was no evidence that gene sharing occurs between the different environments (although this can be implied). As horizontal gene transfer is one of the main routes for bacterial populations to develop AMR, it makes sense to include accessory gene analysis in this study to quantify overlap between clusters and the different sources.

There are also limitations to the sampling as this was not carried out on a single population over a specific sampling period (cross-sectional), meaning that this analysis isn't being tested on a dataset with real transmission events. However, I understand the difficulties in obtaining such datasets. The authors address this briefly in the discussion but Australia is very big and barriers to transmission are likely much more prominent than in smaller geographies. Could the dataset be divided by area and year of sampling and analysis re-run on individual datasets to potentially identify more realistic transmission events?

Minor comments:

Line 27: what kind of linkages? Transmission?

Line 32: what do you mean by clusters?

Line 43: what are the significant challenges?

Line 81: citation for enterobase

Line 81: maybe more on why genomes were used from this study in particular.

Lines 85-86: sentence doesn't make sense. Should 'save' be in there?

Line 103: what model of DNA evolution was used to construct the tree? Jukes-Cantor?

Line 104: why was a cut-off of 40 SNPs chosen?

Line 109: 'Exploratory phylodynamic analysis'. Should this be here?

Line 112: selected 'for' analysis

Line 112: selected for what analysis?

Line 113: could you be more specific when referring to 'this method'

Line 122: typo, should be 'cut-off'

Line 165: why was the threshold of 100 SNPs determined and not higher or lower?

Lines 304-310: could just look within the literature to answer these questions. Are these top STs associated with MDR?

Could also just run Abricate/other AMR software to identify AMR genotypes within these isolates.

In the introduction you mention that your findings serve as a model for other genera (Klebsiella, Salmonella). Would the thresholds for defining clusters differ for these genera compared with your results from this E. coli study?

Figure 2: It is not very clear from the figure which colour key belongs to which metadata around the tree tips. I think this would be better displayed as a rectangular tree with the metadata displayed as separate rows to the right of the tree.

Figure 2 B and C: It isn't clear what the colours on the bar charts represent.

Figure 4: It's quite difficult to see the SNP differences as the thickness of the line becomes lost when stacked on top of each other.

The top cross-source identified clusters also appear to be most abundant in the collection. Could the amount of cross-source connection be possibly due to the fact that there is just more of it being sampled?

Reviewer #2

(Remarks to the Author)

Promoting research to develop better surveillance methods is crucial. Therefore, I was pleased to see this work exploring a relatively neglected and important research area. I agree that more research on SNP thresholds is needed to better define levels of relatedness. However, I have several comments on the methodology:

Can the authors contextualize their proposed approach with outbreak detection? Specifically, how does the proposed method find applicability with cross-species outbreaks recorded in the geographical areas and within the considered period? If this approach is to be considered for surveillance, how does it effectively detect outbreaks and transmission based on the historical data considered in this work?

I am skeptical that "transmission" can be accurately applied here as transmission implies directionality. Can the authors explain why they think their results can truly define transmission also in consideration of the lack of strong temporal/Bayesian evidence?

Can the authors specify how many commensals versus pathogenic E. coli strains were identified per cluster across hosts/years etc?

Additionally, can the authors explain the nature of the SNPs, including their localization, the genes involved, and their positions in the genomes? It would be interesting to understand the functionality of the SNPs and their relevance from a zoonotic perspective.

Did the authors consider potential confounding effects in their workflow?

I agree with the authors that finding novel thresholds to better define what levels of relatedness are appropriate is needed, can the authors provide evidence of the applicability of their methods to different datasets (e.g. E. coli from different geographical regions for which the fixed SNP threshold was used)?

Introduction:

Line 67: Why is combining cgMLST and SNPs a better approach? More details should be provided.

Methods:

Line 80: A descriptive summary of the genomic information is essential. Please specify the number of isolates for each year, the number of pathogens per source/host, etc.

Results:

Line 130: Collection characteristics. Can the authors provide details on the yearly distribution of pathogens/commensals/sources/hosts?

Line 157: Can the authors clarify the rationale for choosing an allelic distance of ≤ 40 to identify clusters of isolates?

Line 155: The statement "Direct or indirect transmission can be inferred by the concurrence of closely related genomes across" implies directionality. Can the authors clarify how transmission can be accurately inferred in this context?

Line 167: Silhouette analysis might not perform well as it can vary depending on the cluster types. Did the authors test other validation indicators? Also, please provide the parameters used for these indicators. Did the authors stratify/analyze the SNP cut-off results in correlation to years/source/collection, etc.?

Line 152: "Utilising a SNP threshold of ≤ 100 rather than ≤ 20 therefore resulted in an 814-fold increase in identification of potential cross-source transmission events, while still requiring isolates to exhibit a high degree of phylogenetic similarity." It seems that the criteria used is the number of clustered isolates to define cross-source transmission. As mentioned above, I

am sceptical that the term "transmission" can be accurately applied here. Can the authors explain the clustering results in the context of geography, in addition to source/hosts? How does the clustering reflect the geography and the years of collections (spatial and temporal transmission)?

Did the authors consider applying their workflow to other E. coli datasets from different geographical areas to understand similarities/differences? And whether their approach has a wider applicability.

Reviewer #3

(Remarks to the Author)

In the Manuscript, "Understanding parameters for One Health genomic surveillance: a retrospective analysis of Escherichia coli from Australia" the authors present an observational study characterizing the genomic differences between pairs of isolates and identifying clusters of multiple versus mono sample sources. The majority of this analysis focused on the 10 most frequently detected STs and characterizing relationships between isolates at different thresholds. These authors perform these analyses on a data set that spans several years and regions of Australia though it faces some limitations because of biases inherent to the data collected. Overall, the manuscript adds to the understanding of E. coli in a One Health context in Australia but faces some limitations.

Major Comments:

1. One major limitation of this study is the lack of epidemiologic information therefore there is no confirmation of exposure to potential outbreak sources. In lines 155-156 suggest using the phrase genomic clusters to emphasize that these are not outbreak clusters.
2. The SKA SNP tool has not been published in a peer reviewed manuscript (reference was to bioRxiv). How did authors validate thresholds to define clusters? Has any validation been performed using outbreak clusters that were identified epidemiologically to support authors stated assumptions of less than 20 SNPs being an epidemiological outbreak cutoff.
3. ST is an imperfect way to group isolates, but often a useful naming tool once isolates are grouped. In some cases, isolates can be within a few SNPs or thousands of SNPs to isolates within the same ST. Isolates can also be in different STs but with relatively few SNPs since you are only looking at pieces of 7 genes. Did the authors investigate other methods to roughly group isolates before performing SKA SNP? Clusters could be defined by cgMLST then SNP analysis used to further investigate differences.
4. Line 173-175, It is surprising that there are so few "clusters" at what is often used as an outbreak threshold, is this due to the wide time span of the samples? Has the SKA SNP workflow been benchmarked with confirmed outbreak data sets to understand the diversity of SNPs seen within a verified outbreak cluster.
5. Line 190, How many isolates comprise a cluster? In the example of ST963 it is the same min,median,max, does this mean there are only 2 isolates in the cluster? It would be helpful to include the number of isolates in addition to the min, median, max. Also consider only include clusters of 3 or more isolates if that is not already defined.
6. Line 214-215, For this section one surprising element the authors do not discuss is time. This is a data set that spans several years and geographic regions and the impact of that is not factored into the analyses for clusters – should that cutoff vary by # of years and location?
7. Paragraph starting at line 223 - Suggest the authors focus on informative distances when doing comparisons between SNPs and alleles (200 or 300 or less SNP differences). Otherwise, there are challenges with doing correlations at higher thresholds, usually SNP based approaches aren't well suited to this, and it is less informative. Also recommend authors describe the inherent biases in the data set that prevent comparison between the two methods. Suggest updating Figure 3 with this information.
8. Line 227 – did the authors investigate why in some cases they found isolate pairs with a low number of allele differences but SNPs into the thousands? Can the increased SNP differences be attributed to accessory genome or intergenic regions? Also for the SNP analysis, how does it account for phage insertion and deletion events which are frequent in E.coli and can cause multiple SNPs associated with one genomic event?
9. Line 266 - Can the authors explain what they mean by spurious metadata? Metadata that is incorrect or incomplete? Not sure how this impacts phylodynamic investigations
10. Line 267 - Can the authors explain how this is novel since there have been other comparisons of cgMLST and kmer based approaches (kSNP in particular) for detecting clusters?
11. Line 273 – Some reference based SNP approaches do pairwise comparisons so this statement is not completely true, not sure what novelty with SKA SNP that the authors are trying to highlight.
12. Line 275-276 - Not sure what the authors are stating here, there are many pangenome, k-mer, MASH based approaches to compare genomes outside of an epidemiologic outbreak investigation. Can the authors better explain their statement and provide references.
13. Line 286 – SKA SNP still had to be subset based on ST - would Pangenome based approaches not work if you subset by ST? What about looking at wgMLST based approaches?
14. Line 336 - Would emphasize that these are genomic clusters and not outbreak or epidemiologic clusters. Pathways for transmission of the strain to humans is unknown because there is no epidemiologic information to identify potential exposures.

Minor comments:

1. Figure 2 – for the tip points indicating Australian states, suggest using a bar since the circles are difficult to see.
2. Figure 4 - Suggest highlighting or explaining in text what counts as a cluster. Suggest labeling clusters with number of SNP differences (min/median/max)
3. Could only find supplemental figure 3A and not 3B - for 3B not clear why FimH was used?
4. Table 1 – Do the cluster size columns refer to the # of samples or the # of SNP differences? The "total" row is confusing since there is not a total for the "cluster size" column. Suggest breaking into 2 tables or removing the "total" for Cluster size

columns.

Reviewer #4

(Remarks to the Author)

Version 1:

Reviewer comments:

Reviewer #1

(Remarks to the Author)

Most of my concerns have been adequately addressed. The methodology is sound and the results presented in the study will be an important addition to the field.

Reviewer #2

(Remarks to the Author)

The authors stated that they considered characterizing the SNPs to be outside the scope of the study. However, in a paper primarily focused on establishing SNP threshold for One Health investigations, providing more detailed information about the SNPs is crucial. This would facilitate better comparisons with other studies that identified small clusters in One Health settings and examined SNPs, helping to understand their functionality and significance, particularly from a zoonotic perspective.

The applicants state that "77.6% (2,298/2,996) of human samples were identified as from clinical sources," but do these samples include both pathogenic and commensal strains? Although distinguishing between commensal and pathogenic *E. coli* lineages may be time-consuming, it can be done through PCR or sequencing. Identifying clusters that include both pathogenic and non-pathogenic strains could be crucial for understanding the composition and significance of these clusters.

Regarding the confounding effects, are the applicants certain that the SNPs could be inflated by the accessory genome? If so, how? Wouldn't it be more plausible for the SNPs to be located in the core genome? Confounding effects could arise from various factors, such as the year or region of sample collection, among others. Did the authors test for these?

Although the authors correctly mention that 'our study is not designed to identify instances of transmission, rather it is to identify which sequence types are observed across multiple sources...are sufficiently closely related to indicate that movement across these sources'. It seems that the aim of the paper is still showing that the use of the proposed methodology is considering transmission in outbreak settings: for example, see lines 106, 291, 324, 337.

The outbreak-like context mentioned by the authors is relevant, and the paper seems to focus on this aspect, given these premises, can the applicant show that their thresholds can truly differentiate an outbreak-context distributed from non-outbreak distributed strains?

I agree with the applicants that considering a more stringent threshold than 100 may be relevant, but why specifically choose 40? Was any testing done to determine if there is a convergence point when adjusting the threshold?

The authors mentioned that: 'We have included five additional indicators (Calinski-Harabasz, Davies-Bouldin, WCSS, Cohesion, and Separation) in our analysis. The complete code for this analysis is available at <https://www.github.com/maxlcummins/APG-OHEC-1>. We see little consensus between these metrics, some favour the 20 SNP threshold and others favouring the 100 SNP with varying statistical significance. As the 20 SNP threshold is not suitable for One Health outbreak investigations e.g. the 20 SNP threshold fails to create cross-source clusters, we have updated the discussion to include a recommendation for the development and validation of a cluster quality indicator that is more suitable for the evaluation of heterogeneous One Health datasets such as those in the present study.'

The authors correctly included five additional indicators to analyze the clusters; however, they report little consensus. It seems the results were not provided. They noted that Ludden et al and Muloi et al did not use tools to assess clusters' validity, and that Murphy et al's methods were unclear regarding whether they analyzed whole dataset core SNPs or pairwise comparisons. However, the methods used in this study for interpreting and validating cluster consistency also fail to provide a clear cluster consistency, raising concerns about how robust/stable are these clusters, especially given the lack of robust temporal Bayesian evidence. In light of the apparent instability of the clusters it's unclear how the authors compare their findings with those of Ludden, Muloi, and Murphy. The authors also claim that the 20 SNP threshold is unsuitable for

One Health outbreak investigations, citing 'The authors state: 'As the 20 SNP threshold is not suitable for One Health outbreak investigations e.g. the 20 SNP threshold fails to create cross-source clusters'. However, in my opinion, the sole absence of 20 SNPs clusters is insufficient to deem it unsuitability for outbreak analysis.

Reviewer #3

(Remarks to the Author)

Authors adequately addressed review comments, manuscript is substantially improved and additional analyses performed.

Reviewer #4

(Remarks to the Author)

REVIEWER COMMENTS

Reviewer #1 (Remarks to the Author):

Summary:

The authors of this study assembled a large and diverse collection of *E. coli* genomes from multiple different sources spanning 36 years in Australia to better understand how the different environments are linked with regards to transmission of *E. coli* isolates. They compared cgMLST and SNP based approaches to cluster the datasets into different groups of related isolates and identified the shared sources they were found in. One Health approaches are gaining a lot of attention with regards to control of AMR and it is important to define transmission of *E. coli* on a large scale using whole genome sequencing.

Major comments:

1. While the authors used a thorough method to define clusters of related isolates, tools for clustering genomes do exist and have been previously used to define clusters in One Health studies. For example, PopPUNK. I'm wondering how the results using this tool would compare to those reported in this study?

Response: We thank the reviewer for their suggestions and constructive feedback.

As a general note, we would like to highlight that we have opted to remove a line in the manuscript detailing the count of novel sequence types we identified during the course of the analysis. Given that the majority of the genomes under study are already published, in addition to Enterobase being unable to assign MLSTs to novel STs for which sequence reads are not available (i.e. the genome assemblies we gathered from Touchon *et al*), we are not able to accurately identify which genomes are of novel sequence types and thus we have chosen to omit this data.

We conducted an analysis using PopPunk as suggested found that all isolates exhibiting 100 SNPs or fewer were of the same PopPunk cluster. This is supported by comparison of PopPunk clustering with cgMLST distances which found strains that differed in their PopPunk clusters had as few as 203 allelic differences; those which shared a PopPunk cluster exhibited as many as 1,184 allelic differences. This latter data-point is similar to that reported by the authors of PopPunk, which found genomes sharing a PopPunk cluster differ by up to 1,000 cgMLST allelic differences. Our SNP clusters (at a threshold of ≤ 100 SNPs) therefore include much more closely related genomes than do PopPunk clusters. We have included this comparison as Supplementary Text 1 and outlined it in the section titled "Identification of STs implicated in cross-source clusters" from line 116.

Our detailed review of the literature identified relevant studies that supported the analytical approach applied in our study and are included in the introduction and discussion sections of the manuscript. For example, network analysis of clusters performed by Ludden *et al.*, (<https://doi.org/10.1128/mbio.02693-18>) utilised core genome SNPs generated from clonal complexed (ST) groups to identify clusters. This method considered samples that differ by 15 SNPs to be related. No specific tool was used for clusters however they do go on to assess a small dataset of 47 ST117 genomes with

BEAST. They use these Bayesian results as support for lack of transmission, however, do not explicitly comment on the dataset appropriateness nor the assessment of a temporal signal using methods such as TempEst or Bayesian Evaluation of Temporal Signal (BETS).

Muloi *et al.*, (<https://doi.org/10.1038/s41564-022-01079-y>) performed a very controlled genomic epi study in peri urban households in Nairobi. They define transmission as 'sharing pairs' isolates that are ≤ 10 cgMLST to indicate strain sharing by direct transmission or common source. Of the isolates that had ≤ 10 cgMLST they validate these by assessing the distribution of 'closely' related pairs as those that differ by < 100 cgSNPs almost all of these differed by fewer than 10 cgSNPs. This study did not break the *E. coli* down into ST groups before performing any core analysis and therefore would have smaller cgMLST and cgSNP distances than those seen in our study. No specific tools were used to assess clusters.

Murphy *et al.*, (<https://doi.org/10.1128/msphere.00738-20>) assessed one health relationships by using SNP clusters that differ by < 50 SNPs, it is unclear if this is whole dataset core SNPs or pairwise. They further assessed "recent" mixing within the B1 phylogroup by using BEAST "To this end, we used the pairwise SNP distance matrix, reconstructed from multiple alignment of genomes mapped to the reference genome". This multiple alignment was of all the *E. coli* strains not broken down by ST and was performed on 128 strains. They assess temporal signal and found a significant R^2 .

2. Although it was evident from this study that different STs/groups of genetically distinct isolates can persist in multiple environments, there was no evidence that gene sharing occurs between the different environments (although this can be implied). As horizontal gene transfer is one of the main routes for bacterial populations to develop AMR, it makes sense to include accessory gene analysis in this study to quantify overlap between clusters and the different sources.

Response: While we appreciate the relevance of this analysis, we feel this is not within the scope of this study and will be presented as a separate study to allow for a more detailed interrogation of such data.

3. There are also limitations to the sampling as this was not carried out on a single population over a specific sampling period (cross-sectional), meaning that this analysis isn't being tested on a dataset with real transmission events. However, I understand the difficulties in obtaining such datasets. The authors address this briefly in the discussion, but Australia is very big and barriers to transmission are likely much more prominent than in smaller geographies. Could the dataset be divided by area and year of sampling and analysis re-run on individual datasets to potentially identify more realistic transmission events?

Response: We subset our collection to a single state and five-year period. Within this time frame, we identified intrasource clusters across 4/10 STs (ST963, ST131, ST95 and ST69) with as few as 40 SNPs, and intersource clusters among 9/10 STs (ST963, ST131, ST95, ST69, ST80, ST648, ST57, ST117 and ST1193) with as few as 0 SNPs. Note however that intrasource clusters could not be observed for STs 57, 95 and 80 as within such STs in this sampling period only a single source was represented. This

is detailed in Supplementary Text and referenced from line 324. We are in the process of performing a prospective snapshot study in a single geographical region in a fixed sampling window to address this limitation of our study design, which we have addressed in the discussion prior but now bolstered in the section titled “Core-genome- and Kmer-based approaches are complementary in large scale One Health genomic surveillance”.

However, in the course of the review process we have also analysed a collection of 1,338 *E. coli* from Kenya by Muloi et al (2022) which includes detailed metadata on the individual households from which *E. coli* sourced from humans and their household livestock. Samples were collected between 2015 and 2016 from 99 households in Nairobi, Kenya (an area of approximately 700km²). This is detailed in Supplementary Text 2 and in the discussion (line 328). In this analysis we identified 207 pairs of strains which differ by 100 or fewer SNPs, some as few as 2 SNPs. Of these, 67/207 pairs were sourced from different host types. 25/67 were collected from the same households, and among the 42 collected from different households, 10 were collected from different households in the same region while 32 were collected from households in different regions. Among strains originating from different host types, we identified examples differing by as few as 4 SNPs. The combination of epidemiological data and the close phylogenetic relatedness of these samples suggests these constitute transmission events. Our data set, being opportunistically aggregated, does not contain such depth of metadata, however this analysis suggests that our approach, when combined with high quality epidemiological metadata, may even be useful in identifying transmission events.

Minor comments:

4. Line 27: what kind of linkages? Transmission?

Response: We have improved the wording of this sentence for clarity that we are meaning genomic linkages (now line 31).

5. Line 32: what do you mean by clusters?

Response: We have added the word ‘genomic clusters’ for improved clarity (line 36). This information is detailed in our methodology (from line 390) and the manuscript more generally.

6. Line 43: what are the significant challenges?

Response: There are many which are described in our 2024 Nature Reviews Genetics piece (<https://www.nature.com/articles/s41576-023-00649-y>), however rather than elaborate here inline we opted to remove this part of the sentence and improve the language inline for clarity.

7. Line 81: citation for enterobase

Response: This reference has now been added line 370).

8. Line 81: maybe more on why genomes were used from this study in particular.

Response: This has now been noted.

9. Lines 85-86: sentence doesn't make sense. Should 'save' be in there?

Response: we have changed 'save' for 'except' in the line in question (line 377).

10. Line 103: what model of DNA evolution was used to construct the tree? Jukes-Cantor?

Response: We used default settings for treebuilding with NJ – the default model for rapidNJ is Kimura rather than Jukes-Cantor. We have clarified on the line in question (line 394).

11. Line 104: why was a cut-off of 40 SNPs chosen?

Response: We assume the reviewer means alleles here rather than SNPs as a 40 allelic distance (and later a 100 SNP distance) was utilised for preliminary screening rather than a 40 SNP distance. Muloi et al (2022) in their "UrbanZoo" publication screened for 'clonal sharing' using a 100 cgMLST threshold and then considered sharing pairs those which share ≤ 10 cgMLST distance, referring to the latter to "indicate evidence of recent strain sharing either by direct transmission or acquisition from a common source". Therefore, we chose a more conservative threshold of 40 cgMLST, rather than 100, as we lack the epidemiological data utilised in Muloi et al and are thus not seeking to detect transmission events. This is now detailed on line 395.

12. Line 109: 'Exploratory phylodynamic analysis'. Should this be here?

Response: This has now been removed.

13. Line 112: selected 'for' analysis

Response: This has been corrected.

14. Line 112: selected for what analysis?

Response: We have added "sequence" to before types to clearly link back to the previous sentence (line 415).

15. Line 113: could you be more specific when referring to 'this method'

Response: We have simplified this sentence for clarity.

16. Line 122: typo, should be 'cut-off'

Response: This has been corrected.

17. Line 165: why was the threshold of 100 SNPs determined and not higher or lower?

Response: Apologies – the line detailing the justification for this was inadvertently removed and has been correctly reintroduced in the methodology section titled “Phylogenetic analysis” (from line 390).

18. Lines 304-310: could just look within the literature to answer these questions. Are these top STs associated with MDR? Could also just run Abricate/other AMR software to identify AMR genotypes within these isolates.

Response: This analysis would be beyond the scope of the current paper, however we are in the process of collating an additional manuscript describing the accessory gene elements identified in this collection.

19. In the introduction you mention that your findings serve as a model for other genera (Klebsiella, Salmonella). Would the thresholds for defining clusters differ for these genera compared with your results from this E. coli study?

Response: We are not suggesting our methodology is directly applicable to other genera, rather that a similar approach may prove useful, so we have improved the language alluding to this for clarity (line 77). It is likely that thresholds suitable for one genus are inappropriate for use in another.

20. Figure 2: It is not very clear from the figure which colour key belongs to which metadata around the tree tips. I think this would be better displayed as a rectangular tree with the metadata displayed as separate rows to the right of the tree.

Response: In our experience this is true for trees with fewer leaves, however given the number of genomes under analysis, to give any degree of interpretability of metadata the tree ends up being far too large and will well exceed the limits of an A4 page. We therefore request to keep the tree in its circular form.

21. Figure 2 B and C: It isn't clear what the colours on the bar charts represent.

Response: This has now been clarified in the figure legend

22. Figure 4: It's quite difficult to see the SNP differences as the thickness of the line becomes lost when stacked on top of each other.

Response: Given the density of some of these networks it is difficult to improve this aspect of the data visualisation. However, to address this, we have included SNP distances in Supplementary Table 2 and noted this in the figure legend.

23. The top cross-source identified clusters also appear to be most abundant in the collection. Could the amount of cross-source connection be possibly due to the fact that there is just more of it being sampled?

Response: This is an important point and certainly possible – we highlight in our discussion that some STs are over-sampled. This is in part due to the relative frequency of some STs within particular niches, and also due to selection biases driven by disproportionate interest by the scientific and clinical communities in strains which

are pathogenic and/or resistant to antimicrobials compared to those which are commensal and/or sensitive to antimicrobials. It is for this reason that we highlight the need for rigorous sampling methodologies in prospective genomic surveillance research in our discussion from line 292.

However, other researchers internationally have identified many of these same sequence types as being potentially implicated in intersource transmission networks. See for example:

- Manges, A. R., & Johnson, J. R. (2012). Food-borne origins of *Escherichia coli* causing extraintestinal infections. *Clinical infectious diseases*, 55(5), 712-719.
 - Details STs 95, 131, 117 and 69
- Saidenberg, A. B. S., Van Vliet, A. H., Stegger, M., Johannesen, T. B., Semmler, T., Cunha, M., ... & Knöbl, T. (2022). Genomic analysis of the zoonotic ST73 lineage containing avian and human extraintestinal pathogenic *Escherichia coli* (ExPEC). *Veterinary microbiology*, 267, 109372.

Reviewer #2 (Remarks to the Author):

Promoting research to develop better surveillance methods is crucial. Therefore, I was pleased to see this work exploring a relatively neglected and important research area. I agree that more research on SNP thresholds is needed to better define levels of relatedness. However, I have several comments on the methodology:

1. Can the authors contextualize their proposed approach with outbreak detection? Specifically, how does the proposed method find applicability with cross-species outbreaks recorded in the geographical areas and within the considered period? If this approach is to be considered for surveillance, how does it effectively detect outbreaks and transmission based on the historical data considered in this work?

Response: SKA is benchmarked on datasets from multiple bacterial species including both simulated and epidemiologically confirmed outbreak scenarios. However, our study is not designed to identify instances of transmission, rather it is to identify which sequence types are observed across multiple sources and which proportion of them are sufficiently closely related to indicate that movement across these sources may take place across broader timelines (and potentially less direct pathways involving a greater number of intermediate hosts and vectors) than within a typical epidemiological 'outbreak scenario'. However, we have in the course of the review process, analysed a cohort over 1,338 *E. coli* genomes from Nairobi. Leveraging the associated metadata and applying a 100 SNP threshold, we found examples of strains which met our cluster criteria (≤ 100 SNPs) which were collected from the same geographical region and in some cases also the same households. This is detailed in the Supplementary Notes and from line 262.

2. I am skeptical that "transmission" can be accurately applied here as transmission implies directionality. Can the authors explain why they think their

results can truly define transmission also in consideration of the lack of strong temporal/Bayesian evidence?

Response: We agree with the reviewer's sentiments despite some of the language we used potentially unintentionally suggesting otherwise. We have removed instances of 'transmission' outside of where a more clearly defined outbreak-like context is being discussed or that of bacterial transmission networks. Throughout the manuscript we also changed 'potential transmission events' to 'strain pairing events' within the context of pairs of closely related genomes analysed using SNP analyses, and limited language of 'potential transmission' to fairly conservative discussion points highlighting the potential for the STs identified to be implicated in intersource transmission, rather than suggesting we have identified bona fide transmission events.

We also added the following in the discussion on line 247 – “Without inferring directionality, our phylogenomic analyses suggest that strains that differ by 50 -100 SNPs are occasionally distributed among diverse hosts or over significant geographic distances, reflecting dissemination pathways that may be complex and poorly understood.”

3. Can the authors specify how many commensals versus pathogenic E. coli strains were identified per cluster across hosts/years etc?

Response: The following line has been added into the results section on line 154: “Where such data was available, 77.6% (2,298/2,996) of human samples were identified as from clinical sources.”

Details among the proportion of human and animal strains from extraintestinal sites and intestinal sites are also mentioned in this same section. Note however that some samples lack detailed metadata pertaining to their specific origin, and faecal/intestinal samples are often not distinguished as being pathogenic or commensal in public databases.

4. Additionally, can the authors explain the nature of the SNPs, including their localization, the genes involved, and their positions in the genomes? It would be interesting to understand the functionality of the SNPs and their relevance from a zoonotic perspective.

Response: While we appreciate the importance of this question, we feel this analysis is outside of the scope of the present study.

5. Did the authors consider potential confounding effects in their workflow?

Response: We have highlighted the potential for SNP values to be inflated by accessory genome elements and elaborated on this in our discussion on line 275 and have also expanded upon the potential confounders introduced due to sampling biases in public databases of pathogenic and resistance from line 330.

6. I agree with the authors that finding novel thresholds to better define what levels of relatedness are appropriate is needed, can the authors provide evidence of the applicability of their methods to different datasets (e.g. E. coli

from different geographical regions for which the fixed SNP threshold was used?

Response: As mentioned, we analysed a collection of 1,338 *E. coli* from Kenya by Muloi et al (2022) which includes detailed metadata on the individual households from which *E. coli* sourced from humans and their household livestock. Samples were collected between 2015 and 2016 from 99 households in Nairobi, Kenya (an area of approximately 700km²). In this analysis we identified 207 pairs of strains which differ by 100 or fewer SNPs, some as few as 2 SNPs. Of these, 67/207 pairs were sourced from different host types. 25/67 were collected from the same households, and among the 42 collected from different households, 10 were collected from different households in the same region while 32 were collected from households in different regions. Among strains originating from different host types, we identified examples differing by as few as 4 SNPs. The combination of epidemiological data and the close phylogenetic relatedness of these samples suggests these constitute transmission events. Our data set, being opportunistically aggregated, does not contain such depth of metadata, however this analysis suggests that our approach, when combined with high quality epidemiological metadata, may be useful in identifying transmission events.

Introduction:

7. Line 67: Why is combining cgMLST and SNPs a better approach? More details should be provided.

Response: This has now been explained from line 70.

Methods:

8. Line 80: A descriptive summary of the genomic information is essential. Please specify the number of isolates for each year, the number of pathogens per source/host, etc.

Response: A detailed breakdown of the dataset is provided in the results from line 82, this includes the source date range and other characteristics. Additional data is available in the supplementary information / supplementary tables.

Results:

9. Line 130: Collection characteristics. Can the authors provide details on the yearly distribution of pathogens/commensals/sources/hosts?

Response: As mentioned above, we have now added information about the clinical origins of human samples to complement the previous descriptions of intestinal/extraintestinal origins within human and livestock hosts. Again, note that our capacity to determine status of commensal and pathogenic *E. coli* lineages which are of intestinal origin from available metadata is limited due to the opportunistic nature of our study.

10. Line 157: Can the authors clarify the rationale for choosing an allelic distance of ≤ 40 to identify clusters of isolates?

Response: Muloi et al (2022) in their “UrbanZoo” publication screened for ‘clonal sharing’ using a 100 cgMSLT threshold and then considered sharing pairs those which share ≤ 10 cgMLST distance, referring to the latter to “indicate evidence of recent strain sharing either by direct transmission or acquisition from a common source”. Therefore, we chose a more conservative threshold of 40 cgMLST, rather than 100, as we lack the epidemiological data utilised in Muloi et al and are thus not seeking to detect transmission events. This is now detailed in the section on phylogenetic analysis from line 390.

11. Line 155: The statement "Direct or indirect transmission can be inferred by the concurrence of closely related genomes across" implies directionality. Can the authors clarify how transmission can be accurately inferred in this context?

Response: We agree this is an important point and have clarified our meaning in this sentence (now line 106), which now reads “Within outbreak settings, direct or indirect transmission can be inferred by the concurrence of closely related genomes across different sources in combination with epidemiologically informative metadata.”

12. Line 167: Silhouette analysis might not perform well as it can vary depending on the cluster types. Did the authors test other validation indicators? Also, please provide the parameters used for these indicators. Did the authors stratify/analyze the SNP cut-off results in correlation to years/source/collection, etc.?

Response: We have included five additional indicators (Calinski-Harabasz, Davies-Bouldin, WCSS, Cohesion, and Separation) in our analysis. The complete code for this analysis is available at <https://www.github.com/maxlcummins/APG-OHEC-1>. We see little consensus between these metrics, some favour the 20 SNP threshold and others favouring the 100 SNP with varying statistical significance. As the 20 SNP threshold is not suitable for One Health outbreak investigations e.g. the 20 SNP threshold fails to create cross-source clusters, we have updated the discussion to include a recommendation for the development and validation of a cluster quality indicator that is more suitable for the evaluation of heterogenous One Health datasets such as those in the present study.

13. Line 152: "Utilising a SNP threshold of ≤ 100 rather than ≤ 20 therefore resulted in an 814-fold increase in identification of potential cross-source transmission events, while still requiring isolates to exhibit a high degree of phylogenetic similarity." It seems that the criteria used is the number of clustered isolates to define cross-source transmission. As mentioned above, I am sceptical that the term "transmission" can be accurately applied here. Can the authors explain the clustering results in the context of geography, in addition to source/hosts? How does the clustering reflect the geography and the years of collections (spatial and temporal transmission)?

Response: Thank you, we agree with this statement. It was not our intention to conclude that these are epidemiologically linked transmission events – we do not have

a sufficiently high-resolution metadata for this purpose and as has been pointed out the geographical and year span of the genomes are broader than appropriate for such analysis. We have changed the language for clarity, instead highlighting an 814-fold increase in identification of cross-source pairs (from line 127).

14. Did the authors consider applying their workflow to other *E. coli* datasets from different geographical areas to understand similarities/differences? And whether their approach has a wider applicability.

Response: As described in responses above, we analysed a collection of 1,338 *E. coli* from Kenya by Muloi et al (2022) which includes detailed metadata on the individual households from which *E. coli* sourced from humans and their household livestock (see Supplementary Text).

A note for Reviewer 2: We would like to highlight that we have opted to remove a line in the manuscript detailing the count of novel sequence types we identified during the course of the analysis. Given that the majority of the genomes under study are already published, in addition to Enterobase being unable to assign MLSTs to novel STs for which sequence reads are not available (i.e. the genome assemblies we gathered from Touchon et al), we are not able to accurately identify which genomes are of novel sequence types and thus we have chosen to omit this data.

Reviewer #3 (Remarks to the Author):

In the Manuscript, “Understanding parameters for One Health genomic surveillance: a retrospective analysis of *Escherichia coli* from Australia” the authors present an observational study characterizing the genomic differences between pairs of isolates and identifying clusters of multiple versus mono sample sources. The majority of this analysis focused on the 10 most frequently detected STs and characterizing relationships between isolates at different thresholds. These authors perform these analyses on a data set that spans several years and regions of Australia though it faces some limitations because of biases inherent to the data collected. Overall, the manuscript adds to the understanding of *E. coli* in a One Health context in Australia but faces some limitations.

Major Comments:

1. One major limitation of this study is the lack of epidemiologic information therefore there is no confirmation of exposure to potential outbreak sources. In lines 155-156 suggest using the phrase genomic clusters to emphasize that these are not outbreak clusters.

Response: We have made additional effort to note the lack of this epidemiological data in the manuscript and made the suggested change throughout the text to highlight this distinction.

2. The SKA SNP tool has not been published in a peer reviewed manuscript (reference was to bioRxiv). How did authors validate thresholds to define

clusters? Has any validation been performed using outbreak clusters that were identified epidemiologically to support authors stated assumptions of less than 20 SNPs being an epidemiological outbreak cutoff.

Response: While this tool is at present unpublished, it has been cited 58 times including at least four occasions within Nature Communications. The manuscript also extensively benchmarks the tool against other methodologies in multiple outbreak data sets of multiple species.

3. ST is an imperfect way to group isolates, but often a useful naming tool once isolates are grouped. In some cases, isolates can be within a few SNPs or thousands of SNPs to isolates within the same ST. Isolates can also be in different STs but with relatively few SNPs since you are only looking at pieces of 7 genes. Did the authors investigate other methods to roughly group isolates before performing SKA SNP? Clusters could be defined by cgMLST then SNP analysis used to further investigate differences.

Response: It is true that strains may be closely related via SNP/allelic distances but differ in their ST. We performed an analysis that found that of the 615 pairs of genomes which differ by ST but exhibit 40 or fewer cgMLST allelic differences. Only 24 pairs differ by 10 or fewer cgMLST alleles, indicating that most of these 615 genome pairs are unlikely to exhibit 100 or fewer SNPs (SNP data is not available for all STs under analysis). In this way our approach may underestimate the occurrence of genomic clusters; we have now acknowledged this in the discussion.

4. Line 173-175, It is surprising that there are so few "clusters" at what is often used as an outbreak threshold, is this due to the wide time span of the samples? Has the SKA SNP workflow been benchmarked with confirmed outbreak data sets to understand the diversity of SNPs seen within a verified outbreak cluster.

Response: Comparatively large time scales and exposure to multiple hosts and environments undoubtedly occur in potential One Health transmission chains. These events expose *E. coli* to diverse biological and physical selection pressures that would be expected to influence mutation frequencies in a manner that drives the accumulation of SNPs compared to what are likely more direct, less protracted and less diverse selection pressures that influence transmissions within single sources. However, we identified 593 pairs of strains across the different STs under investigation which meet a typical outbreak threshold of 20 SNPs. Only three of the pairs included isolates collected from the same source.

SKA has been extensively benchmarked by the authors of the manuscript:

“To test the utility of SKA, it was applied to 7 published datasets from four publications:

- 1) A simulated dataset designed to compare methods for genomic phylogenetics.
- 2) Four benchmarking outbreak datasets for genomic epidemiology, comprising 23 *Campylobacter jejuni*, 9 *Escherichia coli*, 31 *Listeria monocytogenes* and 22 *Salmonella enterica* subsp. *enterica* serovar Bareilly.
- 3) 65 *Staphylococcus aureus* samples from an investigation into an outbreak on a special care baby unit (SCBU) at Addenbrooke’s Hospital, of which 45 have been previously identified as a single outbreak.
- 4) 1,683 ST22 samples from a genomic survey of *S. aureus* from the East of England.”

5. Line 190, How many isolates comprise a cluster? In the example of ST963 it is the same min,median,max, does this mean there are only 2 isolates in the cluster? It would be helpful to include the number of isolates in addition to the min, median, max. Also consider only include clusters of 3 or more isolates if that is not already defined.

Response: This is correct. We have now added the number of clustered genomes per ST to the table, as requested. Regarding the suggested change of clusters from two or more genomes to three or more genomes, this would result in the loss of cross-source pairings in the cluster and table. We appreciate the suggestion but would like to defer to the editor as to the necessity of this change. Regarding inclusion of the count of genomes, this is included in the table in the third column – we have changed the column header for clarity.

6. Line 214-215, For this section one surprising element the authors do not discuss is time. This is a data set that spans several years and geographic regions and the impact of that is not factored into the analyses for clusters – should that cutoff vary by # of years and location?

Response: The cut-off should vary by number of years and location if we were making efforts to identify clusters as being outbreak associated, however given that this is not our intention we feel it appropriate to not adjust for these variables in our capturing of cross-source genome pairings. We have included a breakdown in the supplementary material of source/region/year combinations as well as the clusters identified in a subset of the region/period most well represented in our dataset and refer to it inline. We also highlighted the temporal and geographic span in our discussion and its implications for phylogenomic analysis within the context of the present study.

7. Paragraph starting at line 223 - Suggest the authors focus on informative distances when doing comparisons between SNPs and alleles (200 or 300 or less SNP differences). Otherwise, there are challenges with doing correlations at higher thresholds, usually SNP based approaches aren't well suited to this, and it is less informative. Also recommend authors describe the inherent biases in the data set that prevent comparison between the two methods. Suggest updating Figure 3 with this information.

Response: We agree with the reviewers and have therefore kept our focus on smaller SNP ranges but felt it valuable to provide some context for the spread of SNP ranges throughout ST under analysis. We also agree it is worth highlighting that higher SNP values are less accurate and have noted this in the figure legend. A comment speaking to the comparability of cgMLST and SNP data has also been added from line 291.

8. Line 227 – did the authors investigate why in some cases they found isolate pairs with a low number of allele differences but SNPs into the thousands? Can the increased SNP differences be attributed to accessory genome or intergenic regions? Also for the SNP analysis, how does it account for phage insertion and deletion events which are frequent in *E. coli* and can cause multiple SNPs associated with one genomic event?

Response: We agree with this reviewers' point and acknowledge that large numbers of SNPs may be introduced via the mechanisms outlined and obscure linkage between genomes. However, identifying the location of these SNPs is not practical within the present study. The potential introduction of large numbers of SNPs via lateral gene transfer and recombination events also means that our results are likely a conservative interpretation of the data and that a greater number of matches may be present than what we are reporting – we modified the paragraph structure and expanded upon this topic in lines 214 and 276 of the discussion.

9. Line 266 - Can the authors explain what they mean by spurious metadata? Metadata that is incorrect or incomplete? Not sure how this impacts phylodynamic investigations

Response: This wording has been improved. Line 223 now reads “including but not limited to incomplete metadata associated with large online datasets”. Accurate source labels are important for phylodynamic analyses aiming to provide evidence of cross-source migration events (at a population level, rather than a strain level as is commonly seen to demonstrate transmission events).

10. Line 267 - Can the authors explain how this is novel since there have been other comparisons of cgMLST and kmer based approaches (kSNP in particular) for detecting clusters?

Response: While it is true that other papers have compared cgMLST and kmer based approaches to measuring phylogenetic distance, none to our knowledge feature large One Health datasets. We have therefore adjusted this sentence to highlight that we are speaking to the use of such analysis within a One Health context.

Line 225 now reads “Here we provide novel insight into the intersection of cgMLST-based and SNP-based (SKA) phylogenetic distance measurements within a One Health context to guide future research.”

11. Line 273 – Some reference-based SNP approaches do pairwise comparisons so this statement is not completely true, not sure what novelty with SKA SNP that the authors are trying to highlight.

Response: Reference based approaches typically require the generation of a cohort-wide core genome – SKA-based approaches directly compare each individual genome rather than comparing their similarity as defined through a cohort-derived core genome. It is true that in theory an individual genome could be directly compared to one another in a pairwise fashion without the use of a cohort-derived pangenome, this methodology is not commonplace. However, we have adjusted the line in question to highlight that we are referring to most SNP based methods, rather than all.

Line 228 now reads “This disparity is likely due to the presence of distinct mobile genetic elements between isolate pairs; note that phylogenomic distance using SKA-based methods is performed on a pairwise basis, unlike that of most traditional SNP-based approaches defining a core using pangenome- or reference-based alignments.”

12. Line 275-276 - Not sure what the authors are stating here, there are many pangenome, k-mer, MASH based approaches to compare genomes outside of an epidemiologic outbreak investigation. Can the authors better explain their statement and provide references.

Response: We have adjusted this line to highlight we are referring to a lack of studies within a One Health context specifically.

Line 249 now reads “Knowledge gaps and lack of suitable analytical methodologies for SNP analyses within One Health contexts, as opposed to traditional epidemiological settings, also limits cross-study comparisons. Additional studies are therefore needed to address these gaps in the literature.”

13. Line 286 – SKA SNP still had to be subset based on ST - would Pangenome based approaches not work if you subset by ST? What about looking at wgMLST based approaches?

Response: We agree that there are other methods by which we could preliminarily cluster, such as PopPunk, which we have now performed as an additional approach to clustering. The results of this analysis and its intersection with cgMLST-based clustering have now been included in the supplementary material. While wgMLST has issues relating to scheme management and the identification of novel genetic material, we also now acknowledge the utility of wgMLST-based approaches in the discussion.

14. Line 336 - Would emphasize that these are genomic clusters and not outbreak or epidemiologic clusters. Pathways for transmission of the strain to humans is unknown because there is no epidemiologic information to identify potential exposures.

Response: This has now been clarified by the addition of the following text:

Line 331: Standard clustering metrics were not useful for selecting an appropriate SNP threshold; therefore, we encourage further research on the development and validation of cluster quality indicators that are more suitable for evaluating heterogeneous One Health datasets, such as in the present study.

Minor comments:

15. Figure 2 – for the tip points indicating Australian states, suggest using a bar since the circles are difficult to see.

Response: This change has been implemented.

16. Figure 4 - Suggest highlighting or explaining in text what counts as a cluster. Suggest labeling clusters with number of SNP differences (min/median/max)

Response: We found addition of SNP distance labels to make the figure difficult to view, however we have added SNP distances in Supplementary Table 2 to enable investigation of distances between strains in these clusters for those interested in a more detailed investigation of the data.

17. Could only find supplemental figure 3A and not 3B - for 3B not clear why FimH was used?

Response: Figure 3B is on page two of the Supplementary Figure 3 file. FimH type is used to provide high level within-ST categorisations which form clusters within the figures presented; strains with different FimH types tend to have greater phylogenetic distances than those which share a FimH type. The figure legend has been adjusted to highlight this reasoning.

18. Table 1 – Do the cluster size columns refer to the # of samples or the # of SNP differences? The “total” row is confusing since there is not a total for the “cluster size” column. Suggest breaking into 2 tables or removing the “total” for Custer size columns.

Response: We have removed the “Total” row as requested

A note for Reviewer 3: We would like to highlight that we have opted to remove a line in the manuscript detailing the count of novel sequence types we identified during the course of the analysis. Given that the majority of the genomes under study are already published, in addition to Enterobase being unable to assign MLSTs to novel STs for which sequence reads are not available (i.e. the genome assemblies we gathered from Touchon et al), we are not able to accurately identify which genomes are of novel sequence types and thus we have chosen to omit this data.

Response to reviewer comments

Reviewer #1 (Remarks to the Author):

Most of my concerns have been adequately addressed. The methodology is sound and the results presented in the study will be an important addition to the field.

We thank Reviewer 1 for their feedback and for their efforts in reviewing the article.

Reviewer #2 (Remarks to the Author):

1. The authors stated that they considered characterizing the SNPs to be outside the scope of the study. However, in a paper primarily focused on establishing SNP threshold for One Health investigations, providing more detailed information about the SNPs is crucial. This would facilitate better comparisons with other studies that identified small clusters in One Health settings and examined SNPs, helping to understand their functionality and significance, particularly from a zoonotic perspective.

RESPONSE: We agree that such an analysis would be a useful contribution to the literature but maintain that it is outside of the scope of this manuscript. We are in the process of performing a prospective study that will include E. coli collected from a variety of different sources within a fixed geographical and temporal region, including for example those collected from co-residing companion animals and their owners. This dataset is more appropriate for a qualitative exploration of SNPs and the genes (or intergenic regions) with which they are associated, particularly given the high-quality metadata which will accompany it.

2. The applicants state that "77.6% (2,298/2,996) of human samples were identified as from clinical sources," but do these samples include both pathogenic and commensal strains? Although distinguishing between commensal and pathogenic E. coli lineages may be time-consuming, it can be done through PCR or sequencing. Identifying clusters that include both pathogenic and non-pathogenic strains could be crucial for understanding the composition and significance of these clusters.

RESPONSE: E. coli cannot be reliably categorised as pathogenic or commensal by genotypic means. Efforts have been made to develop genotyping schema to this end (e.g. Johnson et al 2018 [DOI: 10.1093/infdis/jiy459] for the classification of ExPEC with a suite of virulence genes). However, the complex interplay of host factors (e.g. immune status, host species, presence of co-morbidities etc.) and the overlapping role of E. coli virulence-associated genes as fitness factors promoting gastrointestinal colonisation, prevent reliable categorisation of genomes into commensal or pathogenic groups based on gene carriage alone. Future studies leveraging high quality metadata to definitively categorise genomes as commensal- or pathogen-associated, are more appropriate for exploring the potential relationship between virulence, zoonosis and the relevance of genomic clusters of E. coli studied within a One Health Context.

3. **Regarding the confounding effects, are the applicants certain that the SNPs could be inflated by the accessory genome? If so, how? Wouldn't it be more plausible for the SNPs to be located in the core genome? Confounding effects could arise from various factors, such as the year or region of sample collection, among others. Did the authors test for these?**

RESPONSE: Yes, we are certain that SNPs could be inflated by the accessory genome. For example, using SKA to compare two genomes containing identical E. coli chromosomes and either pUTI89 (NC_007941.1) or pCERC4 (NZ_KU578032.1) (two major F-type virulence plasmids in extraintestinal pathogenic E. coli) reveals that they differ by 380 SNPs. F plasmids are common in E. coli, with some of our previous analysis cited on line 237 demonstrating upwards of 75% of E. coli in public databases carrying such plasmid types. While the number of SNPs attributable to mobile genetic elements may vary, mutations in these accessory elements can be included by SKA depending on the degree of nucleotide homology between the accessory elements under analysis.

We are not stating that the SNPs are all located in the accessory genome, only highlighting a limitation of our approach to calculating SNP distances between genomes under analysis.

Detailed analytics and statistical comparisons of geotemporal windows were not performed due to the limited sampling depth; this approach will be more suitable to the prospective study we described earlier.

4. **Although the authors correctly mention that ‘our study is not designed to identify instances of transmission, rather it is to identify which sequence types are observed across multiple sources...are sufficiently closely related to indicate that movement across these sources’. It seems that the aim of the paper is still showing that the use of the proposed methodology is considering transmission in outbreak settings: for example, see lines 106, 291, 324, 337.**

RESPONSE: Line 106 is an objective statement about outbreak level transmissions and the use of phylogenetic distance measurements in combination with epidemiologically useful metadata in their analyses. We then highlight in the next section that we lack such metadata and thus cannot provide the same level of insight, but that SNP data alone may suggest more complex and protracted transmission pathways between sources sharing closely related E. coli. Line 291 highlights this same point and further caveats any potential transmission as taking place ‘... across unknown periods of time via an indeterminable number of intermediate hosts and vectors’. On lines 324 and 337 we also use the word transmission but feel we have made the context clear that we are not speaking to outbreak level transmission.

We have made changes on line 324 and 337 to clarify our meaning and hope that these are to the reviewer’s satisfaction.

Line 324 now reads “By utilising a relatively high threshold for genomic clusters (100 SNP) we account for protracted transmission pathways potentially involving multiple vectors and reservoirs that would expose E. coli to diverse selection pressures⁴⁸⁻⁵⁰, impacting baseline mutation rates⁵⁰ and promoting genetic rearrangements of AMR associated loci⁵¹.”

Line 335-339 now reads “Here we have collected one of the largest E. coli datasets, however, due to the opportunistic sampling and limited metadata associated with this study, our results should be interpreted with the following considerations. Critically, while the current dataset can be used to identify genomic clusters it cannot be used to determine categorically identify transmission events, nor provide insight into the directionality of movement between different sources.”

5. **The outbreak-like context mentioned by the authors is relevant, and the paper seems to focus on this aspect, given these premises, can the applicant show that their thresholds can truly differentiate an outbreak-context distributed from non-outbreak distributed strains?**

RESPONSE: *We are unsure why Reviewer 2 feels we have focused on outbreak-like contexts. We have made every attempt to i) focus our analysis through a lens of a non-outbreak context and ii) contrast throughout the body of the manuscript the requirements for epidemiologically informative, genomics-based outbreak investigations and the lack of metadata available for our study cohort.*

Regarding the second point of this comment, we are not attempting to empirically identify and distinguish phylogenetic distance distributions associated with outbreak-level transmission events and One Health centric non-outbreak, protracted and complex transmission events. We don't feel it is appropriate, or perhaps even possible, to do so without bespoke, large-scale datasets featuring high-quality metadata. Instead, we have sought to perform a phylogenomic analysis with a more relaxed SNP threshold to accommodate potential movements through broader geotemporal windows likely involving multiple intermediate hosts and environments.

6. **I agree with the applicants that considering a more stringent threshold than 100 may be relevant, but why specifically choose 40? Was any testing done to determine if there is a convergence point when adjusting the threshold?**

RESPONSE: *We felt that 40 was a reasonable level of granularity for the present study. Determining a convergence point for distance thresholds presents issues as the resulting convergence will be dataset dependent. Different studies would therefore inevitably use different thresholds, complicating comparisons between research and adding unnecessary methodological complexity to studies in this field.*

7. **The authors mentioned that: ‘We have included five additional indicators (Calinski-Harabasz, Davies- Bouldin, WCSS, Cohesion, and Separation) in our analysis. The complete code for this analysis is available at <https://www.github.com/maxlcummins/APG-OHEC-1>. We see little consensus between these metrics, some favour the 20 SNP threshold and others favouring the 100 SNP with varying statistical significance. As the 20 SNP threshold is not suitable for One Health outbreak investigations e.g. the 20 SNP threshold fails to create cross-source clusters, we have updated the discussion to include a recommendation for the development and validation of a cluster quality indicator that is more suitable for the evaluation of heterogenous One Health datasets such as those in the present study.’**

The authors correctly included five additional indicators to analyze the clusters; however, they report little consensus. It seems the results were not provided. They noted that Ludden et al and Muloi et al did not use tools to assess clusters' validity, and that Murphy et al's methods were unclear regarding whether they analyzed whole dataset core SNPs or pairwise comparisons. However, the methods used in this study for interpreting and validating cluster consistency also fail to provide a clear cluster consistency, raising concerns about how

robust/stable are these clusters, especially given the lack of robust temporal Bayesian evidence. In light of the apparent instability of the clusters it's unclear how the authors compare their findings with those of Ludden, Muloi, and Murphy. The authors also claim that the 20 SNP threshold is unsuitable for One Health outbreak investigations, citing 'The authors state: 'As the 20 SNP threshold is not suitable for One Health outbreak investigations e.g. the 20 SNP threshold fails to create cross-source clusters'. However, in my opinion, the sole absence of 20 SNPs clusters is insufficient to deem it unsuitability for outbreak analysis.

RESPONSE: The results are provided in the supplementary material, however we inadvertently failed to note this in the discussion in our previous submission. This has now been included on line 332.

Throughout our manuscript we have repeatedly acknowledged the noise in our dataset introduced by its opportunistic and retrospective nature. Despite this we feel we have sufficient evidence to highlight the sequence types within Australia associated with multiple sources and exhibiting closely related genomes therein. We also argue that our data convincingly suggests these sequence types may move between these sources, irrespective of whether the data is only sufficient to suggest that this might take place across broader timelines and potentially via multiple intermediate hosts/reservoirs/vectors. We have acknowledged these limitations in the discussion and raise the need for further studies to explore these areas.

In response to "However, in my opinion, the sole absence of 20 SNPs clusters is insufficient to deem it unsuitability for outbreak analysis."; we agree with Reviewer 2. This sentence did not communicate our sentiment effectively. However, the fact that no clusters are observed at this threshold suggests that relationships between genomes from different sources are obfuscated when employing such stringent thresholds that typically involve transmission within a number of days or weeks (as in an outbreak-like event; a context we have reiterated is not broadly applicable to our dataset).

Reviewer #3 (Remarks to the Author):

Authors adequately addressed review comments, manuscript is substantially improved and additional analyses performed.

We thank Reviewer 3 for their feedback and for their efforts in reviewing the article.